# Down-selection of biomolecules to assemble "reverse micelle" with perovskites

Haodong Wu[1,6], Yuchen Hou [1,6], Jungjin Yoon[1,2], Abbey Marie Knoepfel[1], Luyao Zheng[1], Dong Yang[3], Ke Wang[2], Jin Qian[4,5], Shashank Priya [1,2] ✉ & Kai Wang [1,2,4,5] ✉

Biological molecule-semiconductor interfacing has triggered numerous opportunities in applied physics such as bio-assisted data storage and computation, brain-computer interface, and advanced distributed bio-sensing. The introduction of electronics into biological embodiment is being quickly developed as it has great potential in providing adaptivity and improving functionality. Reciprocally, introducing biomaterials into semiconductors to manifest bio-mimetic functionality is impactful in triggering new enhanced mechanisms. In this study, we utilize the vulnerable perovskite semiconductors as a platform to understand if certain types of biomolecules can regulate the lattice and endow a unique mechanism for stabilizing the metastable perovskite lattice. Three tiers of biomolecules have been systematically tested and the results reveal a fundamental mechanism for the formation of a "reverse-micelle" structure. Systematic exploration of a large set of biomolecules led to the discovery of guiding principle for down-selection of biomolecules which extends the classic emulsion theory to this hybrid systems. Results demonstrate that by introducing biomaterials into semiconductors, natural phenomena typically observed in biological systems can also be incorporated into semiconducting crystals, providing a new perspective to engineer existing synthetic materials.

Biomolecules are organic compounds with several types of functional groups such as hydroxyl, methyl, carbonyl, carboxyl, amino, and phosphate groups[1], playing important role in the formation of larger organic structures like proteins, carbohydrates, and lipids. These biomolecules serve as the basic platform for a vast array of activities within the lifecycles of 8.7 million species on Earth, abundantly existing within the simplest unicellular algae to the highly intelligent human being[2]. Exclusively, the chemical origin for the formation of complex cells, tissues, organs and systems, as well as their sophisticated physiological processes (e.g., homeostasis, metabolism), and even more complex life activities (e.g., socialization) are a result of the reactions and interactions among basic biomolecules. Not only biomolecules are responsible for a vast majority of formation and growth of life, but biometal microelements are also actively involved and vital to life[3], which are important across all areas of cellular activities and basic processes such as metabolism and respiration[4]. Interactions between metals and biomolecules can disrupt cellular events when an organism is exposed to, for example, heavy metals (e.g., mercury, lead) or metallic compounds[5]. Investigation of such interaction between metallic compounds and biomolecules in organisms is of great importance, not

[1]Department of Materials Science and Engineering, Pennsylvania State University, University Park, PA 16802, USA. [2]Materials Research Institute, Pennsylvania State University, University Park, PA 16802, USA. [3]Dalian National Laboratory for Clean Energy, Dalian Institute of Chemical Physics, Center of Materials Science and Optoelectronics Engineering, University of Chinese Academy of Sciences, Dalian 116023, China. [4]Huanjiang Laboratory, Zhuji 311800, China. [5]School of Aeronautics and Astronautics, Zhejiang University, Hangzhou 310027, China. [6]These authors contributed equally: Haodong Wu, Yuchen Hou. ✉ e-mail: sup103@psu.edu; kaiwang@psu.edu

only from biomedical perspective, but also for the broader fields of organism-machine interfaces where signals and information can be mutually communicated between the biomolecules and the metallic probes.

Reciprocally, apart from abovementioned introduction of the metal into organic matrix, modulating the metallic compound with biomolecules can also trigger the opportunities for new material discovery and synthesis science (discovery of, e.g., the new metal-organic-framework [MOF][6], metallopolymers[7], and organic-inorganic hybrid perovskites[8], and synthesis of MXenes[9]). Functional innovations in self-healing[10], self-cleaning[11], and self-assembly[12] can be achieved for addressing existing issues within artificial materials. Aligned with this vision we utilize the halide perovskite as a platform and investigate the beneficial role of biomolecules to potentially address the stability and performance issues of perovskite solar cells. Considering the numerous biomolecules that exist in nature, testing the efficacy of all molecules on the perovskites will be arduous and practically impossible. Although material screening can be accelerated by advanced techniques such as machine learning[13], the intrinsic requirement of a source database for training remains missing, requiring the initialized data from both modeling technique and hands-on experiments.

Fundamentally, biomolecules can be classified into finite groups in terms of their functional chemical structures, which are intrinsically responsible for multiple levels of biochemical activities. Prior investigations into the integration of biomolecules with perovskite have delineated a range of mechanisms, from dangling bond[14]/lattice terminal[15]/point-defect[16]/surface passivation[17], to protective barrier introduction[18], crystallization modification[19], and grain boundary[20]/interface engineering[21], etc. However, most of these studies tend to emphasize a singular mechanism, potentially neglecting multiple concurrent and in some cases coupled influences. Practically, mechanisms operating at varied scales can either synergistically or contradictorily interact with each other, leading to versatile holistic manifestations at device level. For instance, parameters like device lifetime, efficiency, and other performance metrics could be differentially impacted by these different mechanisms. This underscores the intricate interplay between the incorporation of an additive and the consequent device performance, which can be mediated through a myriad of mechanisms operating across diverse scales. Hence, a comprehensive understanding, stretching from molecular intricacies to broader grain, interface, and device dimensions, is paramount to decipher the hierarchical impacts of biomolecules on device performance aspects. Adopting such a multi-scale perspective could provide a more in-depth comprehension of the incorporation of biomolecules into perovskite solar cells, directly addressing the central challenges in this field. Considering the extensive variety of biomolecules and the multi-dimensional nature of their impacts, strategic research design coupled with judicious molecular selection becomes essential.

In this work, starting from the molecular perspective, we initiated the study by down-selecting the primitive biomolecules equipped with different functional groups and found that the carbonyl grouped alkyl biomolecules as well as its synthetic derivatives could trigger a self-assembly process forming a reverse micelle-like structure during the perovskite crystallization. This novel structure shows significantly enhanced hydrophobicity and optimally enlarged quasi-Fermi level separation in the device. Using a systematic study including first-principles calculation, massive statistical analysis, microscopic and spectroscopic investigation, a hidden rule for selection of biomolecules has been discovered. Relying on this rule, we demonstrate a paradigm for synthesis of hydrophobically stabilized halide perovskites, i.e., a self-assembled reverse micelle of bio-perovskite. In summary, our work seeks to offer a comprehensive understanding of the influence of various molecular additives on perovskite solar cell performance from molecular scale to module level, providing a useful guide for future research in this field.

## Results
### Down-selection logic for biomolecule additives
In principle, the halide perovskite is a metastable crystal with weakly bonded interaction within their octahedral unit, and the easy atomic lattice-site activation can lead to the emergence of point defects, multilevel traps, and even the lattice collapse to trigger the material degradation particularly in the presence of moisture. Molecular additives in perovskites have shown great efficacy in resolving these issues, e.g., from lattice defect passivation at atomic scale[22], to morphological control[23] and phase stabilization at mesoscale[24], and to interfacial heterojunction engineering at film level[25]. Compared with typical inorganic dopants, biomolecules and biostructures with millions of formulations, could bring new solutions in addressing these issues. Randomly selecting biomolecules to test their effectiveness in perovskites, however, is cumbersome and a long process. By focusing on the specific weakness of the perovskites, i.e., hygroscopicity, and seeking nature-inspired solutions to overcome this weakness may provide the direction towards the rule for material selection.

In natural systems, there are many cellular processes occurring in aqueous environment, which are modulated by the hydrophobic and hydrophilic functional groups within biomolecules. Examples of lipid cell membrane and micelle structure are vital to protect crucial cellular components from dissolution in water. These may also be analogized in perovskite systems if hydrophobic and hydrophilic functional groups are well manipulated. Biomolecules with various functional groups can be a good material pool to be selected from and further introduced into the perovskite solution system. This provides the possibility to down-select a target material that can effectively address all the critical challenges in halide perovskites. To simplify the screening process, the biomolecules were firstly classified into five major classes with respect to their functional groups (Fig. 1a (Tier 1))[26]. Specifically, based on the characteristic functional groups, major biomolecules existing in different forms in biosystems (Supplementary Table 1) can be classified into carbohydrates (monosaccharides, oligosaccharides, polysaccharides), proteins, nucleic acids (deoxyribonucleic acid (DNA), ribonucleic acid (RNA)), and lipids (triglycerides, phospholipids, steroids) with characteristic carboxyl, hydroxyl (and amino), phosphate, and carbonyl (aldehydes and ketones) groups, respectively[27]. Except for these functional groups, the backbone of biomolecules is mainly composed of alkyl hydrocarbons, that is a chain of carbon and hydrogen molecules, intrinsically displaying non-polar features and thus, hydrophobicity. Functionalization of hydrocarbons by such additional chemical groups (i.e., reactive clusters of atoms attached to the backbone) can lead to the reactivity and higher-order subtle interaction with other materials. This triggers the emergence of functionality after complexing with artificial materials such as perovskites. The effect of various biomolecule candidates on perovskites were investigated to conclude a general model of the material that can address the existing issues in perovskites.

To investigate how biomolecules could tune the properties of halide perovskites, representative biomolecules were selected, including niacin, β-estradiol, nicotinamide (NAM), DNA[28], and artemisinin (ART)[29], for each class of biomolecules, as the prototype. Despite previous reports on biomolecules, a comprehensive understanding is missing regarding how a single molecule can elicit varied effects across multiple length scales and how different, yet similar, molecules can produce identical outcomes at these distinct scales. Following this scenario, we intentionally select biomolecules with preference of prior reported ones for the Tier 1 class in order to execute systematic screening over various considerations (widely ranging from mechanisms to performance factors broken down to current, voltage, fill factor, to other device-level factors such as lifetime, cost). This multi-

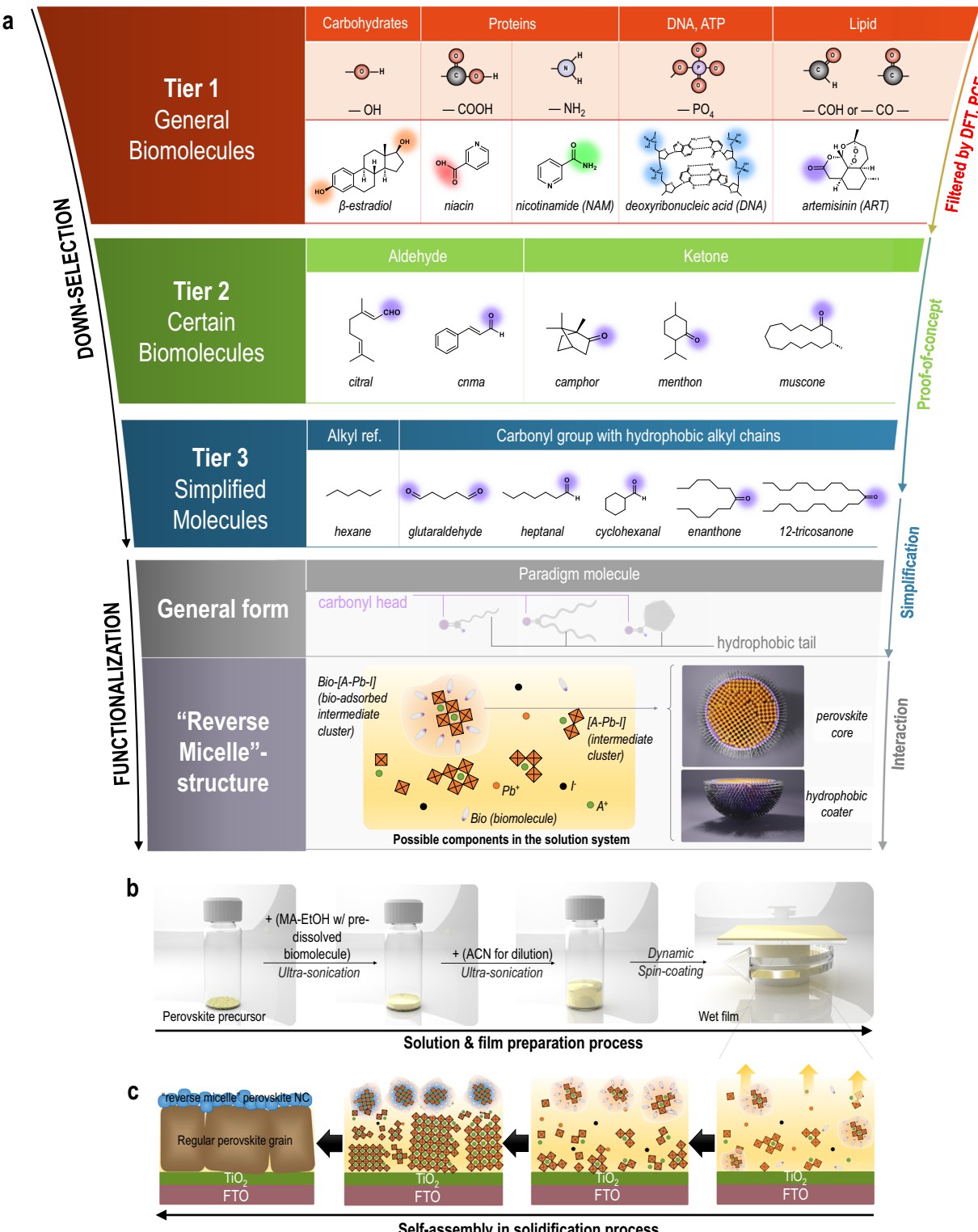

**Fig. 1 | Diagram of down selection rule for the biomolecule additives. a** Down-selection logic: Tier 1 class lists five representative biomolecules (with preference of prior reported molecules) having the characteristic chemical groups belonging to the main five grouped chemical in natural biomolecules, which is systematically screened through a joint consideration on efficiency (broken down into FF, $J_{SC}$, $V_{OC}$), lifetime, cost, etc., to narrow down a candidate towards Tier 2 class. Tier 2 has similar functional groups to the down-selected molecule from Tier 1, and further breaks down the candidate into subgroups of aldehyde and ketone. Evaluation in Tier 2 includes fundamental understanding over full length scale from molecular level, to interface, and device level, with in-depth investigation on photoelectrical and depletion mechanisms as well as the discovery of "reverse micelle". Tier 3 class cross-verifies the concept and up-scales the concept to module level. The general form is concluded to have a carbonyl end group and hydrophobic tails in three different manifestations, coupled with a scheme showing possible components in the solution system. **b** Solution and film preparation process. **c** Film drying schematic showing a self-assembly process to form a bilayer perovskite structure with the top being the "reverse micelle" nanocrystal (NC) and the bottom being the regular grain.

factor systematic screening on the first-tier main grouped materials serves as the initial screening for this study. By comprehensively coordinating the factors for Tier 1 materials, we down-select towards the ART-represented carbonyl family that exhibit the greatest device efficiency and lifetime, and further verify the effectiveness of all the sub-grouped biomolecules with an identical carbonyl group but different carbon backbones (Tier 2 in Fig. 1a), where all the materials exhibit positive contribution to the device, regardless of either aldehyde or ketone type of the carbonyl group. This leads to the hypothesis that amphiphilic molecules in the form of carbonyl hydrocarbon chain (with a carbonyl head and an alkyl tail, schematically shown in Fig. 1) could be the target material. We further modulate these carbonyl biomolecules to verify the assumption in Tier 3 class, with comparisons from alkyl chain molecules with and without multiple carbonyl heads, to verify the hypothesis. It should be noted that beyond biomolecules, we also include synthetic molecules in Tier 3 for comparative study and proof-of-concept purposes. Systematically considering the results, the rule of selection of molecule has been cross-validated to close the loop and a model structure has been provided, leading to the discovery of a "reverse micelle" structure and a bilayer perovskite configuration (detailed in Supplementary Note 5 and Supplementary Fig. 2), which can trigger a series of beneficial effects at molecular level, depletion interface level, charge dynamic level, and all the way towards device and module level. Figure 1b highlights the whole preparation illustration for all the bio-perovskite film processed in this study. The biomolecule-incorporated ink is prepared by a sequential ultrasonication process where the perovskite precursors are firstly dissolved in a mixed solution containing methylamine and biomolecules in ethanol, followed by ultrasonication to form the clear intermediate liquid precursor. Next, extra acetonitrile is added to dilute the ink with a quick ultrasonication, and then directly used for a dynamic spin-coating to form the delicately micro-structured perovskite bilayer (detailed in Methods). Figure 1c shows film drying process where the "reverse micelle" structures are firstly assembled in the solution state, accumulating at the top surface of the wet membrane. Rapid solvent evaporations (due to the low vapor point of solvents, i.e., 82 °C for acetonitrile and 78 °C for ethanol), and synchronized crystallizations occur throughout the film to form two layers of perovskite, with the top layer being "reverse micelle" nanocrystals and the bottom being the regular perovskite grains. This naturally formed structural design originating from spontaneous biomolecule-assisted self-assembly further triggers a series of beneficial effects at different length scales (will be discussed below).

## Primitive screening by Tier 1 biomolecules

Figure 2 shows the detailed investigation of the effectiveness of Tier 1 class on perovskite solar cells. The lattice instability of halide perovskites is the principal issue inhibiting its transition into industrial use. Multiple Lewis acids and bases[30], and other additives capable of cross linkage[31], passivation[17], and terminalization[32], have been introduced to either anchor or passivate the active sites for crystal stabilization due to the molecular interaction between the lattice and the additives. To understand if such a molecular interaction is generally existing between Tier 1 class materials and halide perovskite, first-principles calculations were used to calculate the energy difference as well as density of state (DOS) of active atoms before and after contact. Figure 2a shows the [110] facet of the prototype MAPbI$_3$ perovskite[33] (one of the most commonly exposed facets due to low energy and most frequently observed in high resolution transmission electron microscopy (TEM) images, as shown in Supplementary Fig. 3f). Figure 2b shows the electron density difference, which is defined by the energy difference before and after the adsorption of the biomolecules onto the surface with regard to the individual lattice and biomolecules, obtained from density function theory (DFT) calculation[34]. Generally, all the biomolecules with individual functional groups of carboxyl,

hydroxyl, amino, phosphate, and carbonyl show decreased total energy by −0.38 to −0.91 eV after contact, acting either as an electron donor or acceptor when they come in contact with the partially bonded Pb at the lattice edge, as a result stabilizing the terminal active sites at the lattice boundary. This chemical effect is also verified by the DOS change of atoms from both the perovskite and the biomolecule. For example, Fig. 2c shows the DOS of Pb element from the perovskite and Supplementary Figs. 4–9 compare the DOS changes of both Pb from perovskite and the active atoms from the biomolecules, during the contacting process. The interactions between the perovskite and the four basic nucleotides in DNA[35] (i.e., adenine (A), cytosine (C), guanine (G), and thymine (T), Supplementary Fig. 8) are also calculated, as well as the case of contact formation at other crystal facets (e.g., (100) and (010), Supplementary Fig. 10). Hence, at the molecular scale, attractive interaction between perovskite lattice (terminal Pb with lone electron pair or nonpaired electrons) exists in the cases of all the Tier 1 biomolecules, regardless of the functional groups or the versatile backbone. Nevertheless, this molecular level data may not be sufficient to lead to a conclusive statement on device level.

We utilize these materials into solar cell devices to understand if such molecular level phenomena can also be scaled up to the performance at device level. Figure 2d displays the solar cell device used for the material evaluation, with a classic n-i-p architecture consisting of glass/FTO/c-TiO$_2$/m-TiO$_2$/MAPbI$_3$/Spiro-OMeTAD/Au. We systematically study the influence of five Tier 1 biomolecules on device performance. Supplementary Fig. 11 compares the photovoltaic parameters, including short-circuit current (J$_{SC}$), open-circuit voltage (V$_{OC}$), fill factor (FF), and power conversion efficiency (PCE) of the devices using one of the five Tier 1 biomolecules with concentrations of 0.05, 0.1, 0.5, 1.0 and 5.0 mg mL$^{-1}$ to a reference sample, with a sample capacity of 20 for each group. In general, after exceeding 5 mg mL$^{-1}$, all the biomolecules had negative effects on the device efficiency, which is mainly due to the significant drop in FF (Supplementary Fig. 11). This can be ascribed to the insulative nature of all the biomolecules which hamper the overall charge transfer capability at high concentration and thereby reduce the FF in the solar cells. To understand their real contribution to device performance, we summarize the range of PCE accounting for the best concentration (0.1, 1.0, 0.5, 0.05, 0.1 mg mL$^{-1}$, respectively) for each biomolecule, and compare the best efficiency with respect to different biomolecules in Fig. 2e. The devices with β-estradiol (hydroxyl), DNA (phosphate), and ART (carbonyl) show higher PCE upper limit than that of the device with pristine perovskite. In contrast, the niacin (carboxyl) and NAM (amino) devices show negative and comparable behavior, respectively, compared with the pristine. Since the precursor of perovskite contains methylamine which can have strong neutralization reaction with the carboxyl group in niacin, generating side products such as water into the precursor and additional impurities, there are overall drops in the photovoltaic parameters for niacin incorporated solar cells (Supplementary Fig. 11). While NAM (amino) technically will not induce a neutralization reaction due to its amine nature, it may interfere with the lead iodide reaction with methylamine during the crystallization process. Scaling up from molecular level calculation towards device level performance, besides the aforementioned molecular interactions, other factors such as acid-base reactions and interreference during crystallization can also affect and need to be taken into consideration. Hence, the experimental level investigation will also be crucial in parallel to computational modeling. On the other hand, the primary motivation for utilizing biomolecules with hydrophobic alkyl backbones in perovskites is to understand their effectiveness in strengthening the device stability. Figure 2f compares the aging of normalized PCE of devices (unencapsulated, shelf lifetime) under continuous solar radiation, including the solar heating consideration (~50 °C) and high ambient relative humidity (~65%) which can accelerate the degradation. Under such harsh conditions, devices using

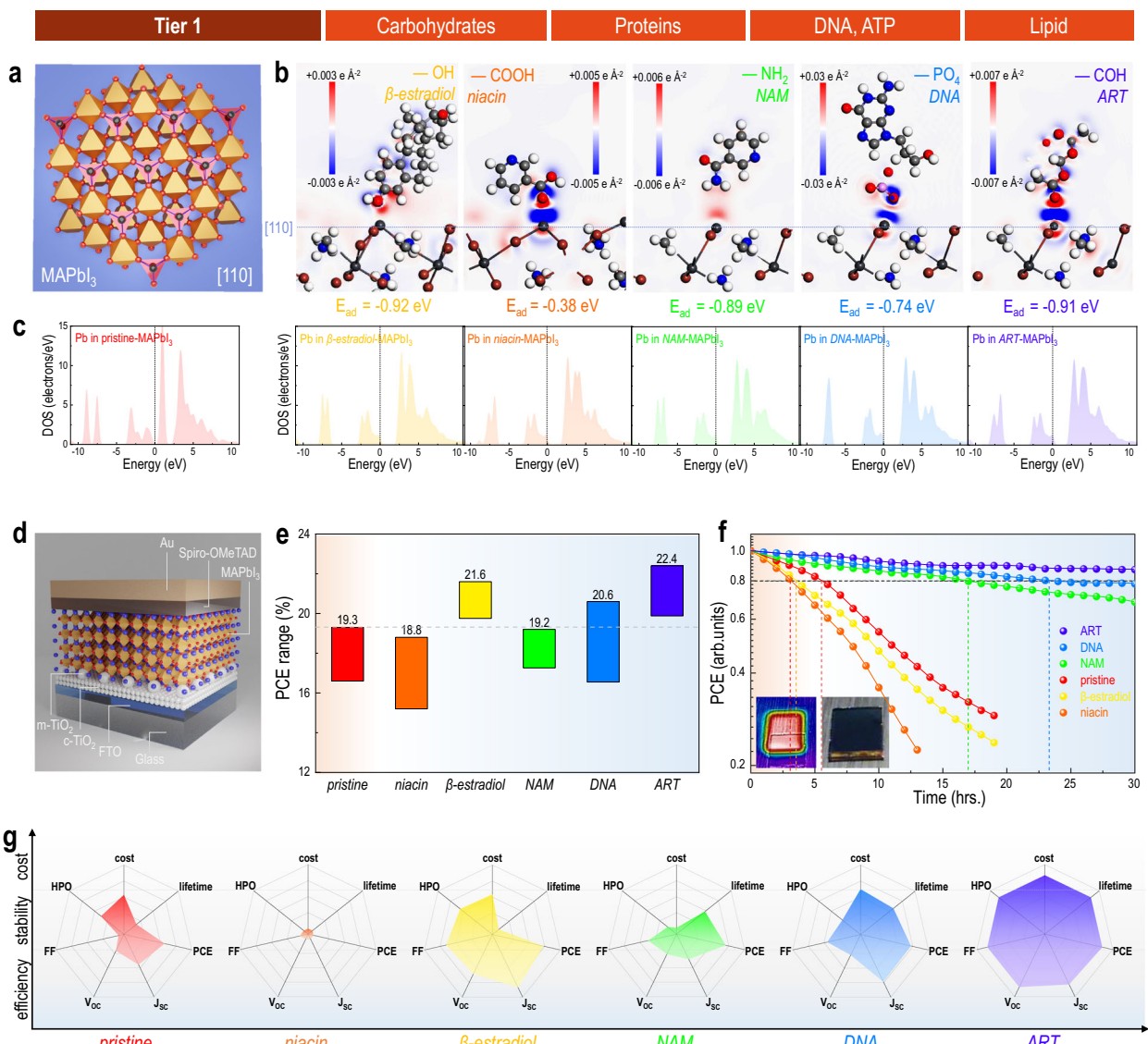

**Fig. 2 | Primitive screening from Tier 1 general biomolecules. a** View of the prototype MAPbI₃ perovskite as a common facet to adsorb biomolecules along the [110] direction. **b** Density function theory (DFT) calculation on the interactions between perovskite and the corresponding biomolecules in Tier 1 class, including niacin, β-estradiol, nicotinamide, deoxyribonucleic acid, artemisinin, along the [110] direction (blue region: accumulated charge density, red region: depleted charge density, E_ad: adsorption energy). **c** Comparison of Density of State (DOS) of Pb between the pristine perovskite and biomolecule adsorbed perovskite. **d** n-i-p device configuration (glass/FTO/c-TiO₂/m-TiO₂/MAPbI₃/Spiro-OMeTAD/Au, FTO: fluorine-doped tin oxide, c: compact, m: mesoporous) used to evaluate Tier 1 class biomolecules. **e** The power conversion efficiency (PCE) range (from minimum to maximum) of corresponding perovskite solar cells. **f** Shelf stability test of unencapsulated perovskite solar cells: devices are exposed in the humid air under 1 sun continuous irradiation (device temperature ~45 °C, relative humidity ~65%), and be tested by I-V characterization periodically. **g** Radar plot for illustration of ability of corresponding biomolecules (under the best concentration) in different aspects, i.e., photovoltaic parameters, lifetime, cost, hydrophobicity (HPO) (detailed in Supplementary Table 2).

NAM, DNA, and ART show 1.56, 1.63, 1.69-fold prolonged lifetimes than the pristine, and significantly better than those with niacin and β-estradiol. We then comprehensively consider the photovoltaic parameters with lifetime, hydrophobicity (specific resistivity to moisture), and cost with relative normalization (Supplementary Table 2), and compare the Tier 1 class biomolecules under the best concentration in the radar plots (Fig. 2g). In general, compared to the pristine, niacin (carboxyl) displays the poorest contribution with regard to all aspects while ART shows the greatest. Other biomolecules either show inferior photovoltaic performance (e.g., not sufficient $V_{OC}$ in DNA), or lifetime (e.g., β-estradiol).

With a large sample capacity for each type of biomolecules, these first rough screening results suggest the biomolecule with alkyl moiety equipped with carbonyl group (e.g., ART) has a positive effect on most aspects of the solar cell. Nevertheless, biomolecules such as ART have a complex backbone structure with additional functional groups such as peroxy-bridge and steric complexity[36], which may dynamically interfere with the carbonyl-lattice interaction when introduced into perovskite matrix. In addition, obtaining such a complex molecule either through extraction from natural species or from artificial synthesis would not be sufficiently cost effective. Further proof-of-concept to verify the hypothesis of the beneficial role of carbonyl functional group with controlled hydrocarbon backbones is needed along with the understanding of the specific mechanisms at microstructural level. Additional experiments were designed to answer the question whether there exists a general molecular structure that can provide unique microstructure for new stabilization mechanism in halide perovskite matrix.

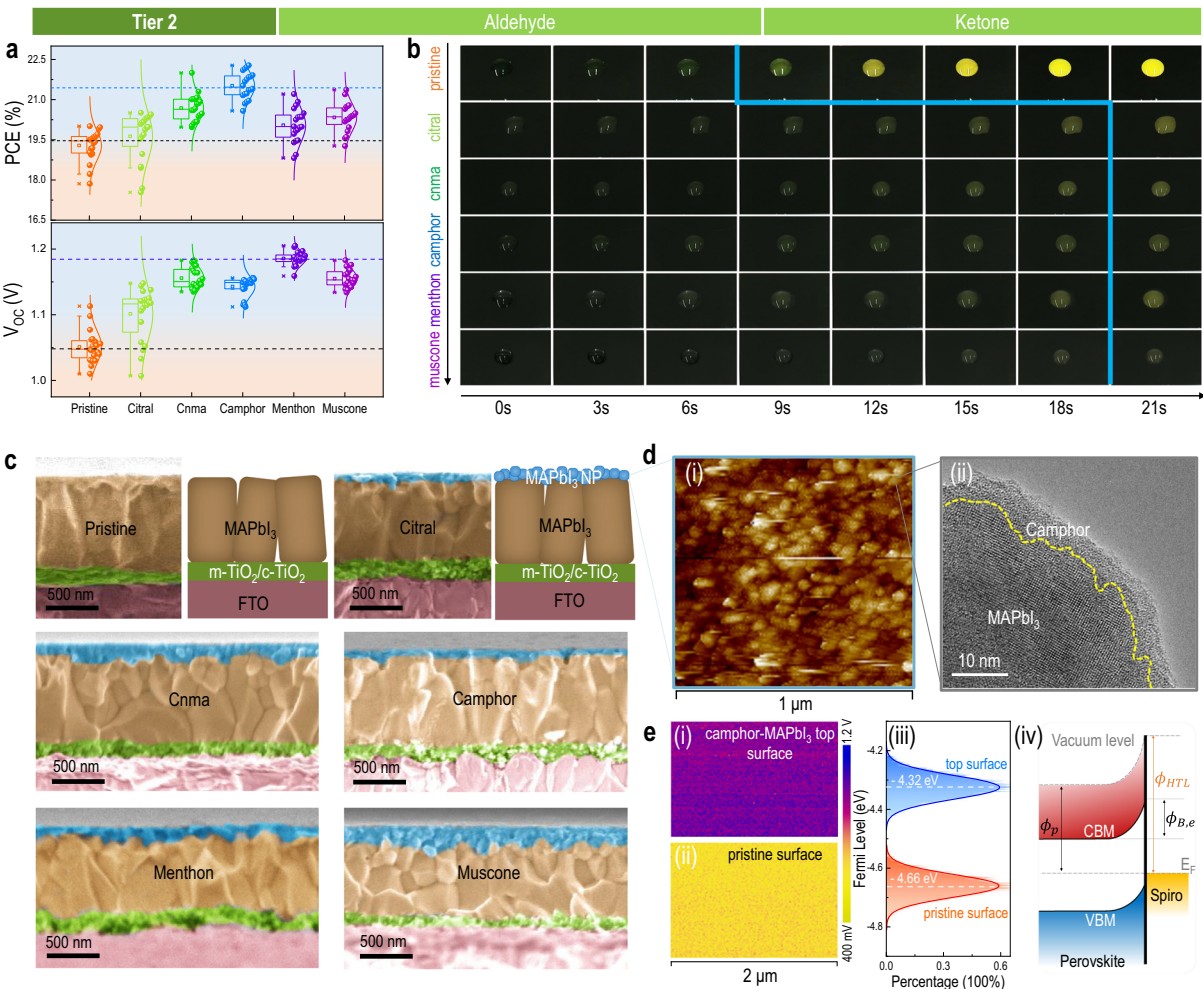

**Fig. 3 | Proof-of-concept by Tier 2 biomolecules. a** Statistical analysis on PCE and $V_{OC}$ from solar cells (20 devices for each group) using Tier 2 class materials, including citral, cinnamaldehyde, camphor, menthon, and muscone, with comparison to the pristine. In the box plot, upper – means maximum value, upper × means 99th percentile, box upper edge and – means 75th percentile and 95th percentile, middle □ means mean, middle – means median, box lower edge and – means 25th percentile and 5th percentile, lower × means 1th percentile, lower – means minimum value. A slight amount of Cl is doped into all the MA-based perovskites to obtain higher efficiency in parallel to test the generality of the biomolecular effect on this Cl-doped perovskite. **b** Film waterproof test by directly dropping 10 μL water on top of different perovskite films in the ambient condition. **c** Cross-sectional SEM images and schemes comparing the structural difference between pristine and biomolecule incorporated perovskites. The biomolecule

doped perovskite exhibits a stacking layer configuration with a large crystal bottom layer and a nanoparticle top layer. **d** Characterization of the nanoparticle top layer: (i) topographical AFM image confirming the nanoparticle feature of the top layer, displaying the particle dimension of ca. 45 nm; (ii) TEM image showing the microscopic configuration of the particle where the perovskite crystal is wrapped by an ultrathin biomolecular coat. **e** KPFM results of surface potential mapping for both (i) biomolecule incorporated perovskite and (ii) the pristine reference. (iii) Fermi-level distribution of different perovskites, extracted from contact potential difference (CPD) (detailed in Supplementary Note 1 and Supplementary Fig. 17d). (iv) Schematic of the energy level bending at the interface of perovskite layer and Spiro-OMeTAD HTL, revealing the Schottky barrier for reversed electron transfer across this interface.

## Proof-of-concept via Tier 2 biomolecules

To do so, we move forward to the Tier 2 class materials. As shown in Supplementary Fig. 12, carbonyl group biomolecules can be further sub-grouped into functional groups of aldehydes (e.g., (geranialdehyde) citral, (cinnamaldehyde) cnma) and ketones (e.g., (2-camphanone) camphor, (L-menthan-3-one) menthon, (moschusketone) muscone). It should be noted that here we utilized 1% Cl doping in MAPbI$_3$ to obtain higher device performance as well as to test the biomolecular effect on this high-performance platform. We tested both subgroup biomolecules using the abovementioned device architecture. It is found that all the Tier 2 biomolecules exhibit overall improved PCE, where the increase is mainly due to the increase in $V_{OC}$ (Fig. 3a) compared to FF and $J_{SC}$ (Supplementary Figs. 12b–e, Supplementary Table 3). Notably, by using menthon, a maximum $V_{OC}$ of 1.21 V was observed, significantly higher than typical threshold of 1.13 V for MAPbI$_3$ solar cells[37] and approaches the theoretical limit of 1.32 V

under AM 1.5 G for this material[38]. The effect of camphor doping concentration on device performance is shown in Supplementary Fig. 19 (photovoltaic parameters can be find in Supplementary Table 4), in which the 3 mg mL$^{-1}$ can give the best efficiency. Figure 3b breaks down video clips of water droplet on pristine and Tier 2 class biomolecules-incorporated perovskite films. The pristine film exhibits a quick color change of the water droplet starting at 9 s, where the yellowish color can be understood by the decomposed PbI$_2$ dissolved into the water. In contrast, all the Tier 2 class biomolecules stabilize the perovskite film well, as can be observed in the longer time for the color change to occur at approximately 21 s. This can be understood by the biomolecule-assisted surface hydrophobicity enhancement as evidenced by the result of enlarged water contact angles from 22° to 64° in Supplementary Fig. 13. To microscopically understand the origin of waterproofing, we employ scanning electron microscope (SEM) to visualize the cross-section of these biomolecules-incorporated

perovskites. As seen in Fig. 3c, the pristine perovskite device displays a regular multilayer device configuration with a single layer of perovskite consisting of large columnar grains connecting both electron and hole transfer layers at bottom and top, respectively. In contrast, after adding Tier 2 biomolecules, all the samples show a bi-layer structure of perovskite with a bottom layer of regular large columnar grains but a thin top layer of nanoparticles. Nevertheless, the X-ray diffraction (XRD) results display identical spectra for all the samples (Supplementary Fig. 14a), suggesting no amorphous or new phases emerged, and that the top nanoparticle layer is also in *Pm3m* (space group) of MAPbI$_3$, identical to that of the bottom layer. Taking camphor-MAPbI$_3$ as an example, we employ high-resolution SEM and transmission electron microscope (TEM) with elemental mapping to investigate the nature of the nanoparticles in the top layer. Supplementary Fig. 14b shows a top view of the nanoparticle layer, consisting of closely packed nanoparticles with dimension of ca. 45 nm. We take out one nanoparticle and visualize the cross-section structure using high-resolution TEM, and find that the nanoparticle consists of perovskite lattice in the core and a coating of amorphous organics outside (Supplementary Figs. 15d–f, where the scanning-TEM (STEM) and fast Fourier transform (FFT) results verifies its pseudo cubic nature in the core with a [110] plane exposed outside for contact formation with organic shell), revealing a "reverse-micelle" structure. This perovskite-biomolecule reverse-micelle configuration is further schematically shown in Supplementary Fig. 16c, and confirmed by high-angle annular dark-field (HAADF) in Supplementary Fig. 16e, where the Pb and I elements exclusively from perovskite are distributed within the core region, whereas the C element that can be either from MA in perovskite or the functional groups in the biomolecules pervade over the whole region. Overall, the formation of such nanoparticle structure is analogous to the reverse micelle structure of amphiphilic molecules. All of the Tier 2 biomolecules used here have a hydrophobic tail (alkyl moiety), and a carbonyl head (either aldehyde or ketone) that can bond with perovskite due to the predicted attractive interaction from aforementioned DFT results of Tier 1 biomolecules (it should be noted other groups may also form the interaction while here our focus is on the branch of carbonyl group due to the joint consideration of other factors in Fig. 2g). Such an assumption will be further verified in Tier 3 class molecules and formation mechanisms will be discussed later.

Thanks to this interesting microstructure, we also observed a significant V$_{OC}$ improvement for the device when a layer of reverse-micelle nanoparticle is interfacing with bottom bulk perovskite and the top-hole transfer layer (HTL). The V$_{OC}$ in solar cell is defined by the separation of quasi-Fermi level of electrons and holes under nonequilibrium conditions such as illumination[39]. The separation degree determines the V$_{OC}$ value which is fundamentally limited by the bandgap, electronic band structure of the active layer as well as the interface contacting with the transfer layers modulated by the losses to recombination[40]. It should be noted that the molecular interaction between perovskite and the biomolecule can adjust the DOS of atoms in perovskite near the surface regions, adjusting the electron deficiency by donor-acceptor reaction or sharing with the biomolecule. The reduced nanoparticle size down to nanometer scale can also trigger quantum confinement effect and quantize the energy band[41]. These can lead to the changes in the electronic properties, particularly changing the Fermi-level (significantly affecting the charge behavior across the interface, depletion region, and thus solar cell efficiency) of the perovskite-biomolecule reverse micelle nanoparticle. We use the kelvin probe force microscope (KPFM) to firstly quantify the Fermi-level of the nanoparticle layer. Figure 3d shows the nanoparticle film topography with corresponding TEM feature of one particle, and Fig. 3e displays the surface potential mapping results (detailed in Supplementary Note 1). The pristine MAPbI$_3$ film displays a Fermi level of −4.66 eV consistent with the typical reference value, while the camphor-MAPbI$_3$ displays a value of −4.32 eV (340 meV upwards

toward conduction band). Such a higher level can be beneficial to the selective charge transfer across the perovskite/HTL interface. Briefly, at this interface due to the band bending of both conduction band minimum (CBM) and valence band maximum (VBM) near the interface as well as the low Fermi level of a typical HTL (e.g., −5.22 eV of doped Spiro-OMeTAD), there is an ohmic contact for holes but a Schottky barrier for electrons, respectively. Ohmic contact for holes facilitates holes extraction and Schottky barrier for electrons benefits the hole-only transfer via blocking electron diffusion. How efficiently the electrons are blocked can be quantified by the height ($\Phi_{B,e}$) of this Schottky barrier, by

$$\Phi_{B,e} = \Phi_{HTL} - \Phi_p \tag{1}$$

where $\Phi_{HTL}$ and $\Phi_p$ are the work function of HTL and perovskite, respectively. The higher Fermi level of perovskites will lead to a larger barrier for unwanted electron reversion transfer. Such a blocking effect also preserves in nonequilibrium condition such as illumination (detailed in Supplementary Fig. 18 and Supplementary Note 2), where the $\Phi_p$ simply needs to be adjusted to quasi-Fermi level of electrons ($\Phi_{p,e}$) (which will be specifically discussed later). As a result, the nanoparticle perovskites have a larger electron barrier of 900 meV at the perovskite/HTL interface, compared to 560 meV for pristine MAPbI$_3$, which is supposed to provide a more efficient electron blocking effect at the anode and thus result in less recombination losses.

To verify the reduced losses of recombination, we further investigate the recombination process of different solar cell devices. The light current density-voltage (J-V) curve in Fig. 4a(i) highlights a noticeable improvement in V$_{OC}$ when comparing the pristine device with the camphor-MAPbI$_3$ device. This improvement reflects a concurrent rise in V$_{OC}$ (around 10%) and FF when incorporating the biomolecule. In essence, the FF reflects the charge transport feature in device, particularly predominant at smaller internal fields. Figure 4a(ii) manifests the comparison in this perspective. By plotting photocurrent ($J_{ph} = J_i - J_d$) vs. effective bias ($V_{eff} = V_0 - V$) (where $J_i$ is the current density under illumination, $J_d$ is the dark current, $V_0$ is the voltage when $J_{ph} = 0$, and $V$ is the applied bias voltage, respectively)[37], we observed higher $J_{ph}$ in the camphor-MAPbI$_3$ device under lower $V_{eff}$ region (<0.15 V), suggesting a more efficient charge transport behavior in such device (detailed explanation is incorporated in Supplementary Note 3). In addition, light-dependent investigations have been conducted for both devices under both short- and open-circuit conditions. As seen in the light-dependent V$_{OC}$ plot in Fig. 4a(iii), the less deviated slope of 1.72 $kT/e$ from camphor-MAPbI$_3$ device than that of 1.95 $kT/e$ from the pristine device with regard to the ideal case (1 $kT/e$), suggests reduced trap-assisted recombination[42] (Shockley-Read-Hall, SRH recombination) in the camphor-MAPbI$_3$ device. Consistently, Fig. 4a(iv) shows the power-law fitting of the log-log plot of J$_{SC}$ vs. light intensity, smaller deviation of the power index in the camphor-MAPbI$_3$ device ($\alpha = 0.94$) than that of pristine device ($\alpha = 0.88$) with perspective to the ideal case ($\alpha = 1.00$), reveals a minimized recombination losses (detailed explanation is incorporated in Supplementary Note 3).

In addition to the electric results discussed above, a deeper understanding of the photophysical process is equally important. It should be noted that simply utilizing PL study to reveal the charge extraction efficiency is unrigorous. As seen in Fig. 4b(i), photoexcitation can lead to Shockley-Read-Hall (SRH) recombination losses, PL emission, and extraction by HTL. Observation of PL intensity change remains inadequate to conclude the ratio between SRH losses and extraction by HTL, where the former is harmful to device performance but the latter is beneficial. In real device working conditions, due to the presence of internal field (asymmetric electrodes), the charge transfer across the perovskite/HTL interface is dominated by a drifting process

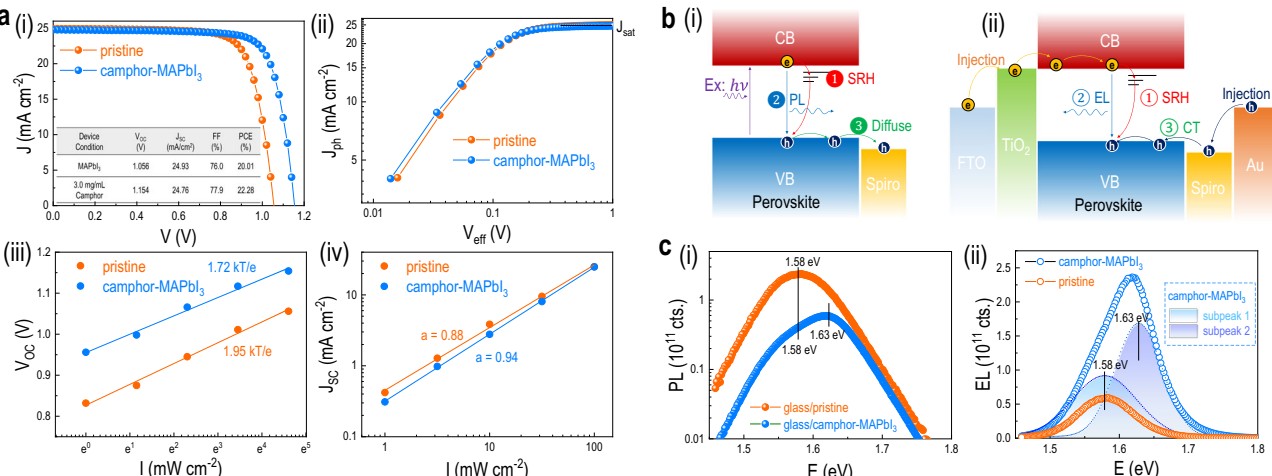

**Fig. 4 | Fundamental (photo-) electronic verification of improved device performance. a** Recombination study on the large $V_{OC}$ explanation: (i) J-V curve and corresponding photovoltaic parameters, (ii) photocurrent density-bias voltage ($J_{ph}$-$V_{eff}$) characteristic, (iii) open circuit voltage-light intensity ($V_{OC}$-I) characteristic, and (iv) Short circuit current-light intensity ($J_{SC}$-I) characteristic of different devices. **b** Explanation on the higher potential of the photocurrent by photoluminescence (PL) and electroluminescence (EL) studies: schematics of (i) PL and (ii) EL mechanism for illustrating the incomplete information extracted only from PL results (Ex excitation, CB conduction band, VB valence band, SRH Shockley-Read-Hall recombination, CT charge transfer). **c** Corresponding results of (i) PL spectra of perovskite/HTL bilayer and (ii) EL spectra of complete solar cell device.

rather than a simple diffusion, which will make the above model less applicable for analyzing these real cases. Thus, additional information is needed to identify the respective contributions from both SRH losses and extraction. Here we introduce the supplementary investigation by electroluminescence (EL) using the solar cell device but reverse the charge flow by current injection. As schematized in Fig. 4b(ii), in condition of an identical current injection, the injected electrons can have three pathways of SRH recombination losses, electroluminescence (bimolecular recombination) emission, and losses during charge transfer (CT) across multiple interfaces and within layers. Both losses to SRH recombination and CT are negatively contributing to the device. Thus, compared to PL, the result from EL can be a more direct index to evaluate the device performance. We carry out both PL and EL investigations for devices using either pristine or camphor-MAPbI₃ perovskite. Figure 4c(i) shows the PL results of samples consisting of perovskite with a top Spiro-OMeTAD HTL. The pristine sample displays a spectrum with a peak around 1.58 eV which is consistent to the bandgap of MAPbI₃. In contrast, the camphor-MAPbI₃ perovskite displays a blue shift (Supplementary Fig. 20b(ii)), exhibiting a peak at 1.63 eV, accompanied by lower states peaked at 1.58 eV. The larger bandgap of 1.63 eV is most likely belong to the particle-like perovskites in the upper layer (Supplementary Fig. 14c) while the lower states (1.58 eV) are consistent to the bottom columnar grains identical to the pristine. As mentioned above, albeit the camphor-MAPbI₃ perovskite displays lower PL intensity, it is difficult to distinguish the contribution from either SRH recombination or charge extraction. Similarly, transient PL (Supplementary Fig. 20b(iii)) with information of photocarrier lifetime also cannot distinguish either. While the EL results in Fig. 4c(ii) reveal a clearly conclusive result. The camphor-MAPbI₃ device exhibits a 5-fold higher EL intensity than the pristine one, suggesting reduced losses through SRH or CT or both, which is consistent with the results of recombination studies. Similarly, two sub-peaks were observed which are centered at 1.58 and 1.63 eV, respectively, from the camphor-MAPbI₃ device (Supplementary Fig. 20c(iv)). This is consistent to the PL results and suggests that besides the conduction band frontier states (corresponding to 1.58 eV), there is an additional excited state of electrons at an energy level 50 meV higher (corresponding to 1.63 eV) (Supplementary Fig. 20c(v)). The higher energy state can also contribute to a higher potential energy to the excited electrons, jointly with the reduced

recombination losses, leading to the higher $V_{OC}$ of the devices with Tier 2 class biomolecules.

## Further validation via Tier 3 molecules

So far, we have verified the improved efficiency and hydrophobicity of the devices fabricated with biomolecules containing carbonyl head and alkyl moieties. Nevertheless, the question of whether the alkyl tail needs to be a complex moiety or with steric extension (e.g., camphor, menthon) or just a simple carbon backbone remains to be answered. Further molecular simplification and generalization would be of importance to the application level for practical usage and cost consideration. To answer this question, the Tier 3 class molecules (Supplementary Fig. 21) are investigated. As specified in the inset of Fig. 5a, three types of molecules were used, namely hydrophobic (i.e., hexane (hex), alkyl chain only), hydrophilic (i.e., glutaraldehyde (glu), alky chain with two carbonyl end groups), and amphiphilic with one carbonyl group and different alky tails (where the tail is of different lengths and structures, including dihexyl ketone (dih, one carbonyl head with two hexyl tails), cyclohexanecarboxaldehyde (cyc, one carbonyl head with one cyclohexane tail), heptaldehyde (hep, one carbonyl head with one hexyl tail), and laurone (lau, one carbonyl head with two dodecane tails)). According to the abovementioned hypothesis that the reverse micelle particle layer originated from the amphiphilic biomolecules, it is expected that the hex or glu would not give rise to such a structure and result in insufficient device performance, while the rest of the molecules (dih, cyc, hep, and lau) would render the reverse micelle and result in higher stability and efficiency in the solar cell devices. Figure 5a compares the PCE and $V_{OC}$ of small area solar cells using each of the Tier 3 materials (with representative J-V characteristics shown in Supplementary Fig. 21c (photovoltaic parameters in Supplementary Table 5), and full parameter statistics in Supplementary Fig. 22). As expected, all the devices with simplified amphiphilic molecules (dih, cyc, hep, and lau) regardless of the length, geometry, or spatial extension, display overall enhancement in PCE, particularly the $V_{OC}$ with a general increase of 8.2% from 1.056 to 1.143 V, compared to that of the reference solar cells. In parallel, both devices with hydrophobic (hex) or hydrophilic (glu) molecules display inferior performance with even lower $V_{OC}$ values averaging at 1.03 V and 0.91 V, respectively. To further correlate the efficiency with the hypothetical micelle particle structure, we randomly select the hep

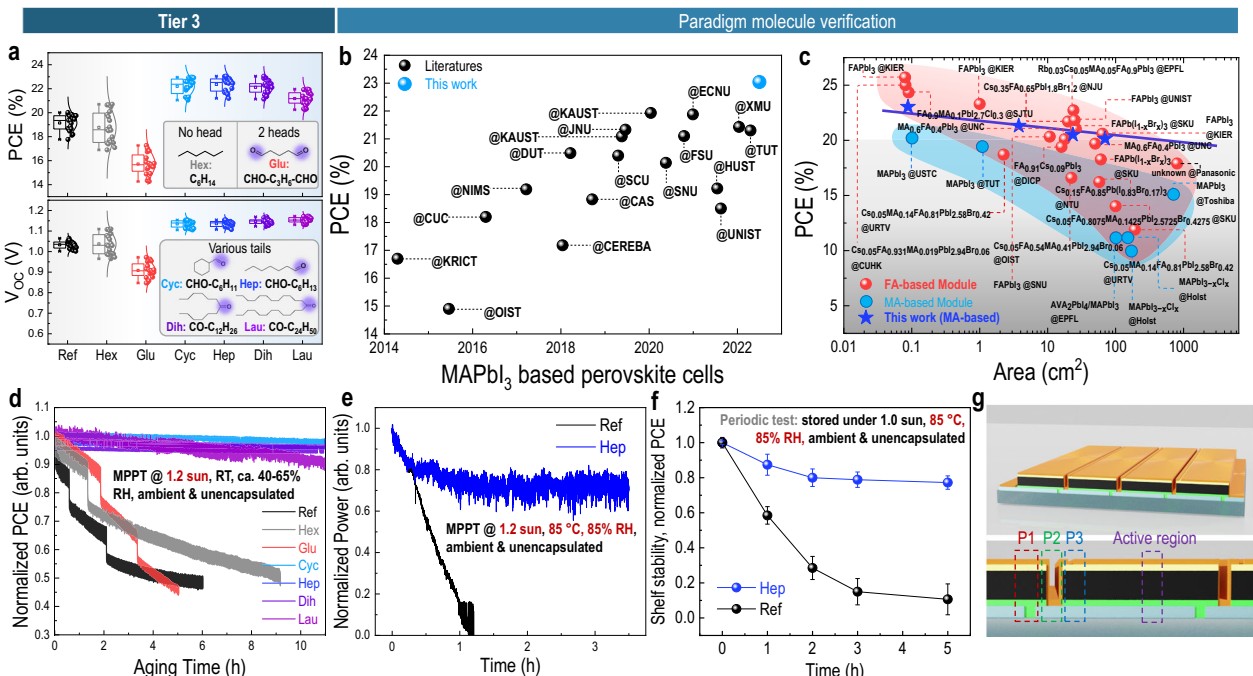

**Fig. 5 | Sub-carbonyl biomolecule investigation via Tier 3 class molecules and validation on scaling up to module level application. a** Statistic analysis on PCE and $V_{OC}$ of perovskite solar cells (Ref: reference/pristine) with corresponding additives, including hexane (hexH, no carbonyl head), glutaraldehyde (glutaral, 2 carbonyl heads), cyclohexanecarboxaldehyde (cyclohexanal, 1 head and 1 cyclic tail), heptaldehyde (heptanal, 1 head and 1 linear tail), dihexyl ketone (7-tridecanone, 1 head and 2 linear tails), diundecyl ketone (laurone, 1 head and 2 long tails). **b** PCE comparison among MAPbI$_3$-based solar cells, data is collected from prior literature since 2014 which is detailed in Supplementary Note 4 and Supplementary Fig. 1a. **c** Up-scalability of different perovskite modules, via PCE vs. device active area plot. Data is collected from prior literature detailed in Supplementary Fig. 1b. Red dots (red region) correspond to FA or mixed complex composition of perovskite, blue dots (blue region) correspond to MA-based perovskite with a simple MAPbI$_3$ formula. The star mark denotes the results from this work, showing great improvement of MA-based perovskite compared with prior results by using the bio-

molecule and investigate the hep-concentration dependent device behavior and film morphology. Supplementary Fig. 23a, b compare the J-V curves (photovoltaic parameters in Supplementary Table 6) and cross-sectional SEM images of devices with hep of different concentrations. Along with the increase of hep concentration from 0 to 5 mg mL$^{-1}$, the particle top layer from SEM gradually increases in thickness from 0 to 195 nm. At a concentration of 3 mg mL$^{-1}$, a champion PCE of 23.04% has been obtained with a large $V_{OC}$ of 1.148 V. Notably, when compared with other MAPbI$_3$-based solar cells from the literature, this work reports one of the highest values for the MAPbI$_3$-based perovskite solar cells to date (Fig. 5b). Albeit FA or other mixed A- or X-site perovskite compositions have been quickly advanced to higher performance, the learning from this study with successful demonstration in the classic MAPbI$_3$ composition may also provide fundamental insights to the broader community.

One example could be the module level demonstration, with regard to the up-scalability that is one of the major issues of the perovskite photovoltaics. Here our first attempt is on the 3.76 cm$^2$ minimodule device by incorporating all the Tier 3 class molecules with constant concentration of 3 mg mL$^{-1}$ (Supplementary Fig. 23d, Supplementary Table 7). The minimodule is designed to have four strip sub-cells (Supplementary Fig. 23e) with a width of ca. 5 mm for each from the joint consideration of both dead area and sheet resistance of the lateral conduction in FTO. Consistent with the small cell results, all the minimodules with amphiphilic molecules (dih, cyc, hep, and lau)

assisted reverse micelle strategy, and indicate good competition potential of these bio-MA-based perovskite among state-of-the-art FA-based perovskites. (In addition, the PCE survey of perovskite solar cells with previously reported biomolecule additives[48–50] is detailed in Supplementary Fig. 1c). Maximum power point tracking (MPPT) of unencapsulated minimodules under **d** High illumination intensity (i.e., 1.2 sun, room temperature (20–25 °C), and ambient environment (circa (ca.) 40–65% RH)), and **e** high intensity of illumination, humidity and temperature (i.e., 1.2 sun, 85% RH, and 85 °C). **f** Shelf-stability test of corresponding minimodules exposing under 1.0 sun illumination at high humidity and temperature (85% RH and 85 °C), data is collected from I-V characteristics measured at 1$^{st}$ h, 2$^{nd}$ h, 3$^{rd}$ h, 5$^{th}$ h (error bar indicates mean value ±standard deviation). Shelf-stability of encapsulated device exhibiting T99 exceeding 3000 h is included in Supplementary Fig. 27. **g** Schematic of module layout for illustrating micro-connection at P1, P2, and P3 regions.

display overall enhanced PCE compared to those either from pristine, hydrophobic (hex) or hydrophilic (glu) ones. We also optimize the molar concentration of the molecule (using hep as an example) for module devices with different dimensions to evaluate the up-scalability of this amphiphilic molecular strategy. Supplementary Fig. 24 displays the J-V curves and photovoltaic parameters of devices with area scaling from 0.088 cm$^2$ (single cell), to 3.76 cm$^2$ (4-strip minimodule), to 23.5 cm$^2$ (10-strip small module), and to 70 cm$^2$ (18-strip module). All the devices display the PCE over 20% regardless of the device sizes, and significantly higher than prior reported values for MAPbI$_3$-based modules (~200% higher than prior highest at ~100 cm$^2$ scale, Fig. 5c). Although it has been widely agreed that FA-based perovskites typically demonstrate higher efficiency than the MA-based perovskites, here we observe that by using the amphiphilic strategy, the MA-based perovskites can also achieve the efficiency levels of those FA-based modules, and exhibit two-fold greater up-scalability when considering the PCE drop over the increasing area (i.e., estimated slope in Fig. 5c, −0.0306 from molecule-MA-based devices of this study vs. −0.068 from the state-of-the-art FA-based devices).

Overall, we have down-selected general biomolecules towards a simplified amphiphilic model, by using hep as an example to verify the reverse micelle concept all the way to the module-level. This leads to the remaining question of the effect of simplified molecules on module-level lifetime under various stimuli, as well as how the degradation mechanism works at this level. It should be noted that to reflect

the degradation during working conditions, we employed the dynamic maximum power point tracking (MPPT) test (rather than the shelf storing-testing cycle) for all the module devices. Figure 5d shows an accelerated aging test of various fresh minimodules without encapsulation, under 1.2 sun illumination in ambient condition (room temperature and a relative humidity (RH) ranging from 40 to 65% recorded by a RH meter), by a dynamic MPPT instrument (where a locally fluctuating bias is applied to the device and an output current is being tracked to calculate the power, the sudden drop of the curve in Fig. 5d is due to the auto-selection of the bias fluctuation range which is preprogrammed in the software). All devices using amphiphilic molecules (dih, cyc, hep, and lau) display significantly improved stability compared to the pristine perovskite reference device. Although hex and glu show slightly prolonged lifetime compared to the reference, their performance still quickly drops to 50% of the initial performance, while the dih, cyc, hep, and lau devices shows negligible performance drop (<5% drop) after 11 h. Importantly, although the sample stage is isothermally controlled to be at room temperature (20–25 °C), the solar thermal effect still can quickly heat up the device to a surface temperature approaching ~50 °C as observed in Supplementary Fig. 25. Nevertheless, taking these issues into consideration, dih, cyc, hep, and lau devices still exhibit robust resistance to the degradation. We further increase both RH and temperature to 85% and 85 °C, respectively, in the ambient environment for unencapsulated module, according to the industrial photovoltaic aging standards (ISOS-D-3)[43]. As seen in Fig. 5e, taking the hep minimodule as an example with reference to the pristine device, the reference device exhibits a linear PCE drop to zero within 1 h, while the hep minimodule stabilized to 70% of its initial value after 3 h. Similarly, we also test the shelf stability of the devices (continuously 1.0 sun illumination but periodically tested J-V) and observe greater stability from the hep minimodule (Fig. 5f and Supplementary Fig. 26). All the devices were tested in the ambient environment without encapsulation. Although the aging of hep minimodule is orders of magnitude slower than the reference device, the absolute lifetime of these unencapsulated devices are of insufficiency. To peer with real usage condition, we then use atomic layer deposition (ALD) to encapsulate the module device with further capping of a glass substrate, and test the lifetime (Supplementary Fig. 27), which shows significantly improved lifetime. In specific, negligible PCE drop (<1%) from the hep minimodule lasting for 3 kh has been observed compared to the ca. 2 kh 50% drop in the reference device. These results reveal that encapsulation can be of great importance for practical use, while at molecular scale, modification within the perovskite layer can be the secondary factor determining the overall life span.

There have been many prior studies investigating the degradation mechanisms but mostly for small area devices. Shifting to the modules, both the micro-series connection between neighboring strip cells and the P1, P2, and P3 scribed structures can be affected by the aging, particularly considering the perovskite morphological evolution during aging and its effect on these microstructures remains unclear. This, however, could be important in understanding the module degradation mechanism. To understand this, we employ the cross-sectional SEM to monitor the structural changes upon aging at local positions of P1, P2, P3 and active regions (Fig. 5g), with/without a specific type of biomolecule (hep). SEM images for pristine and hep modules were measured at both fresh and 3 h aged conditions (continuously illuminated under 1 sun at 85% RH and 85 °C). The pristine module shows the presence of pores and dramatic morphological change of the perovskite after 3 h aging. The appearance of these pores can peel off the contact, increase series resistance, and cause the active sites to develop cracks and fissures. The pores are spread out over the whole area of the pristine perovskite layer, which indicates the degradation mechanism is triggered by a general decay spreading over the whole perovskite layer upon aging. This suggests that the micro-connection

at the dead area of P1, P2, P3 regions may not be the major driving force for device degradation, instead, it is the perovskite layer itself that initiates and develops the degradation. Therefore, stabilizing the perovskite layer is critical even for the module level stability, which can be achieved by the reverse micelle strategy. As seen in Fig. 6, the hep-perovskite shows great stability where there is no noticeable morphological change among all the regions after 3 h aging under illumination. This can be ascribed to the hydrophobization of the reverse micelle layer formed with the assistance of the amphiphilic molecules.

In principle, this "reverse micelle" design can be further added onto more stable baseline perovskites such as FA-based perovskite. While in order to form this metastable colloidal-crystallization system, it is necessary to balance the multi-interactions in the solution so that the colloidal crystallization can proceed along the micelle-kinetic pathway, which will need careful consideration dealing with the precursor concentration, solvent selection, biomolecular solvation degree, etc. Here we present an example in the Cs-dope FA-MA system (Supplementary Fig. 29), where through carefully tuning the solution parameters, a bilayer perovskite with the top being "reverse micelle" nanocrystals has been successfully obtained. And the consequent multiple positive effects on both efficiency and stability are well validated as well.

## Discussion

Overall, through a systematic study over a broad material inventory, we have discovered a rule of down-selection of certain biomolecules can trigger the formation of "reverse micelle" structure in the colloidal-crystallization metastable system. Here we provide the summary of this work: (1) The down-selection study originates from the ambiguous, contradictory, and incomplete understanding of how biomolecular additives can positively improve the perovskites at different length scales (revealing the hidden framework linking the molecule to the device); (2) 15 molecules pre-grouped in the Tier-1, 2, and 3 lead to a hierarchical screening, towards a down-selected molecular template that can lead to new colloidal-crystallization dynamics converging to a discovery of "reverse micelle" structure; (3) systematic studies from molecular scale (Å to nm), to meso-scale (e.g., grain, μm), interface scale (e.g., interfacial depletion, mm), small cell and module scale (cm), reveal the advantageous "reverse micelle" structure in solving critical issues in perovskite photovoltaics. Specifically, (i) we discover that "reverse micelle" nanocrystals bearing higher Fermi level and higher energy state, will not only block the unwanted reverse transfer effect but also enlarge the splitting of the quasi-Fermi level under illumination, leading to a higher $V_{OC}$ (towards 1.2 V) that approaches the limit of the MAPbI$_3$ perovskite. (ii) On the perspective of lifetime, such hydrophobic reverse micelle significantly resists moisture penetration and enhances the device stability under various testing conditions via MPPT, shelf-stability testing, with and without encapsulation, etc. In a larger scope, this work presents an example to develop innovative perovskite colloidal manifestations with unexpected merits for applications.

## Methods

### Materials

Moschusketone (Muscone, 97%), Cinnamaldehyde (Cnma, 99%), Geranialdehyde (Citral, 95%), 2-Camphanone (Camphor, 96%), L-Menthan-3-one (Menthon, 98%), Diundecyl ketone (Laurone, 95%) was purchased from Fisher Scientific. Lead (II) chloride (PbCl$_2$, 99.99%), Bis(trifluoromethane)sulfonimide lithium salt (Li-TSFI, 99.95%), anhydrous chlorobenzene (CB, 99.8%), methylamine solution (MA, 33 wt.% in absolute ethanol), 4-tert-butylpyridine (4-TBP), Titanium(IV) isopropoxide (TTIP, 97%), anhydrous acetonitrile (ACN, 99.8%), hexane (hexH, 99%), glutaraldehyde (glutaral, 70%), β-estradiol (β-E, 98%), Dihexyl ketone (7-Tridecanone, 97%), Heptaldehyde (Heptanal, 95%), Cyclohexanecarboxaldehyde (Cyclohexanal, 97%), Artemisinin (ART,

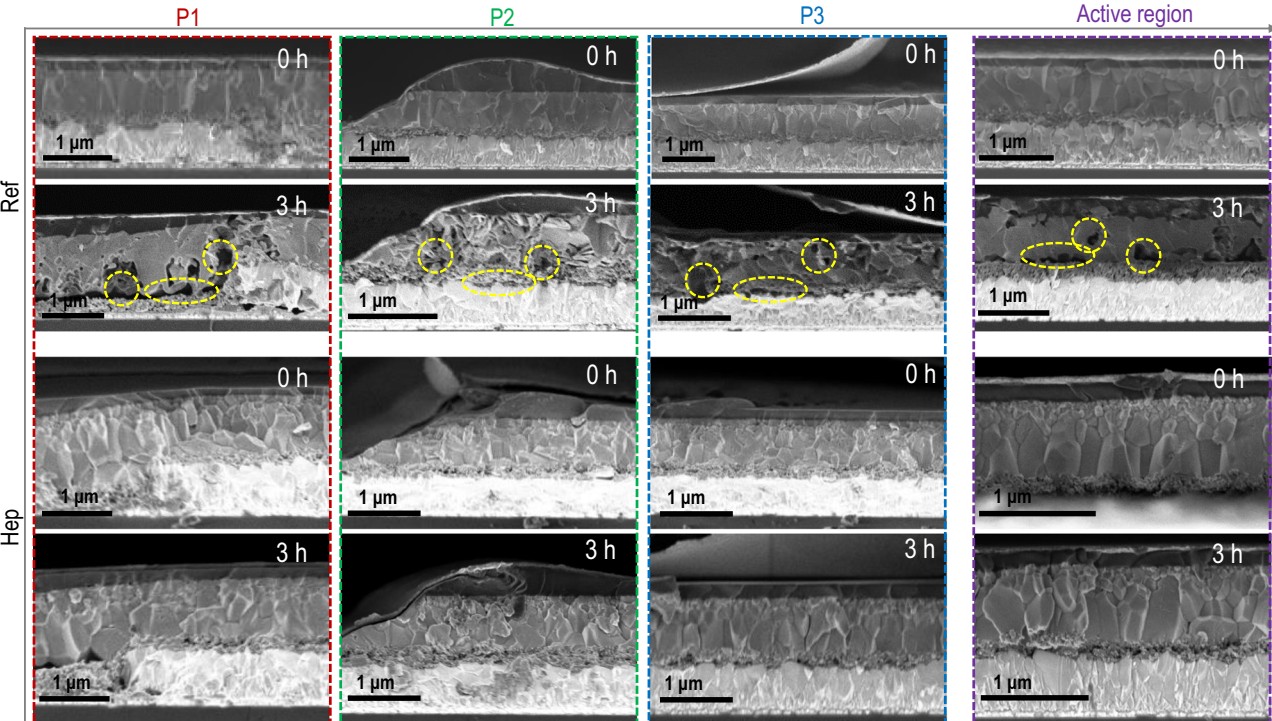

**Fig. 6 | Degradation investigation of module devices based on Tier 3 class molecules.** Degradation mechanisms by SEM measurement showing the micro-feature of different minimodules at conditions of initial and after 3 h aging (being kept under 1.0 sun, 85 °C and 85% RH for 3 h). Specific regions P1, P2, P3, and active regions are compared for each group (detailed in Supplementary Fig. 28).

98%), Nicotinamide (NAM, 99.5%), Niacin (VB-3, 99.5%), Deoxyribonucleic acid sodium salt from testes (DNA, with molecular mass of $1.3 \times 10^6$ Da (~2000 bp), %G-C content of 42.1%), Hexadecyl trimethyl ammonium chloride (CTMA, 98.0%), Poly(triaryl amine) (PTAA, average $M_n$ 7000–10,000), anhydrous toluene (99.8%), fullerene (C60, 99.5%), Bathocuproine (BCP, 96%) were purchased from Sigma-Aldrich. Lead iodide (PbI$_2$, 99.9985% metals basis) was purchased from Alfa Aesar. Methylammonium iodide (CH$_3$NH$_3$I (MAI)), Titania paste (18NR-T) was purchased from Greatcell Solar Materials. 2,2',7,7'-Tetrakis[N,N-di(4-methoxyphenyl)amino]−9,9'-spirobi-fluorene (Spiro-OMeTAD, 99.5%) was purchased from Luminescence Technology Corp. FTO glass (sheet resistance of 7 Ω/sq) and ITO glass (sheet resistance of 8 Ω/sq) were purchased from Wuhan Jingge Technology Corp. Gold (Au, 99.99%) and silver (Ag, 99.999%) were purchased from Angstrom Engineering Inc. All materials were used as acquired without further processing.

**Preparation of perovskite solution with biomolecules**
The perovskite solution was prepared according to our previous work[29,44,45]. Take 3 mg mL$^{-1}$ heptanal MAPbI$_3$ perovskite solution as an example: at the beginning, 6 mg heptanal was added to 1 mL methylamine solution (MA, 33 wt.% in absolute ethanol) to obtain the 6 mg mL$^{-1}$ heptanal-MA solution. Then 500 μL of the 6 mg mL$^{-1}$ heptanal solution was added to a vial containing 159 mg MAI and 461 mg PbI$_2$ powders. The vial was directly sonicated for 1 h until the solution became completely clear. It should be noted the stocked methylamine solution (MA, 33 wt.% in absolute ethanol) may have a lower concentration than 33 wt.% due to the evaporation of MA from ethanol. It is highly recommended to use fresh and new methylamine solution. After obtaining a clear solution from the abovementioned sonication, 500 μL ACN solvent was added to dilute the system. After a quick sonication for 5 min, the final 3 mg mL$^{-1}$ heptanal MAPbI$_3$ perovskite solution was ready to use within 1 day. Perovskite solutions with other biomolecules were prepared with the same process.

**Fabrication of small-area perovskite solar cell**
Small-area perovskite solar cells (active area: 0.088 cm$^2$) with a n-i-p structure (glass/FTO/c-TiO$_2$/m-TiO$_2$/MAPbI$_3$/Spiro-OMeTAD/Au) were fabricated according to our previous work[28,29]. The fluorine-doped tin oxide (FTO) substrates were pre-rinsed with detergent, DI water, acetone, and isopropanol in the ultrasonic cleaning bath sequentially. Then the substrates were dried in an oven. Before casting, the front sides of FTO substrates were processed by 30 min UV/Ozone treatment. In a fume hood, the c-TiO$_2$ was spin-coated onto the FTO substrate at a spin speed of 2000 rpm for 30 s by dropping the diluted TTIP solution (TTIP:ethanol = 1:10, volume ratio), followed by annealing at 150 °C for 30 min. Then the m-TiO$_2$ layer was spin-coated onto the c-TiO$_2$ layer at a spin speed of 6000 rpm for 30 s by dropping the TiO$_2$ paste solution (TiO$_2$ paste:ethanol = 1:5, weight ratio), followed by annealing at 120 °C for 30 min. Then the substrate was transferred into a glovebox, the perovskite layer was spin-coated on the TiO$_2$ layer at a spin speed of 4000 rpm for 30 s without annealing. During the perovskite crystallization process, the bi-layer characteristic (top nano-crystalline capping layer/bottom typical large perovskite grains) are self-assembled. Next, the spiro-OMeTAD layer was spin-coated onto the perovskite layer with the same recipe as perovskite spin-coating by using the premade spiro-OMeTAD solution (72 mg spiro-OMeTAD dissolved into 1 mL of CB with 26 μL of 4-TBP and 13 μL 520 mg mL$^{-1}$ Li-TSFI-ACN). Then the substrate was moved into ambient environment and an 80 nm Au was deposited onto the spiro-OMeTAD layer by thermal deposition in the instrument chamber at a pressure of $1 \times 10^{-6}$ hPa. Briefly, the p-i-n structure perovskite solar cells (ITO/PTAA/MAPbI$_3$/C60/BCP/Ag) were fabricated as follows: the ITO/glass substrate was precleaned as mentioned above. The PTAA layer was spin-coated onto the ITO layer using the premade PTAA solution (2 mg mL$^{-1}$ in toluene) at 6000 rpm for 30 s with annealing process at 100 °C for 10 min. The perovskite layer was spin-coated onto the PTAA layer as mentioned above. Then a 15 nm C60, 8 nm BCP, and 100 nm Ag was deposited onto the perovskite layer as mentioned above.

## Fabrication of large-area perovskite module

A pico-second (ps) UV laser scribing machine (OpTek Systems MM2500, 355 nm) was employed to fabricate perovskite solar modules with different geometric designs, the interconnection creation process (P1, P2, P3) is similar to that used in CIGS devices[46]. In general, the P1 scribing process was performed on the pre-cleaned FTO substrates (of different sizes including $3 \times 3 \, cm^2$, $6 \times 6 \, cm^2$ and $10 \times 10 \, cm^2$, which consist of 4, 10 and 18 sub-cells with dimension $0.47 \times 2 \, cm^2$, $0.47 \times 5 \, cm^2$ and $0.45 \times 9 \, cm^2$ respectively, the corresponding device active area (excluding dead area) is $3.76 \, cm^2$, $23.5 \, cm^2$ and $70 \, cm^2$, as shown in photos in Supplementary Fig. 24) to separate the bottom FTO electrode. In general, the individual strip cell was controlled to have a width ca. 5 mm, by considering both the sheet resistance of FTO and processing complexity. The following layers of c-$TiO_2$/m-$TiO_2$/$MAPbI_3$/Spiro-OMeTAD were spin-coated onto the P1-processed FTO substrate, using conditions identical to those described in the fabrication of small-area perovskite solar cell. It should be noted that the perovskite ink was slightly diluted to slow down the crystallization and the ink was filtered by 0.45 μm filter to keep a homogeneous film feature. The P2 scribing process was performed on the abovementioned semi-finished devices, in order to create contacting area between the FTO and Au, as well as separate each sub-cell and establish electrical channel. Parameters including laser power, laser frequency, moving speed, were carefully tuned to minimize the thermal damage on these layers but keep the channels clean from impurities and structural damage. After this, a 60 nm Au layer was deposited onto the premade substrate, followed by the P3 scribing process to separate the Au electrode. During the P3 process, we used low intensity laser power to simply separate the Au layer whilst keeping the bottom layer continuous. This is to avoid device shortage issues, otherwise the Au may melt and be attached to bottom FTO layer. For the encapsulated device, ALD (Kurt J. Lesker ALD150LE Cluster Tool) deposition of Parylene C and $Al_2O_3$ were utilized, followed by a glass capping with the assistance of UV curable resin (Norland Optical Adhesive No. 81), which is specified in Supplementary Fig. 27b.

## Thin film, precursor solution and device characterization

As for thin film characterization, scanning electron microscopy (SEM) measurement was conducted by a field-emission SEM instrument (Zeiss Merlin LEO 1530). Transmission electron microscopy (TEM) measurement was conducted by a dual aberration-corrected TEM instrument (FEI Titan3 G2 60–300). Energy dispersive X-ray spectrum (EDS) measurement was conducted by a SuperX EDS system attached to TEM instrument. Kelvin probe force microscopy (KPFM) measurement was conducted by a Bruker Innova atomic force microscope (AFM), in which the tip potential calibration was based on a standard highly oriented pyrolytic graphite (HOPG) and the surface potential measurement probe was Pt/Ir coated conductive probe (SCM-PIT-V2). Steady state photoluminescence and time-resolved photoluminescence (PL and TRPL) were conducted by a fluorescence spectrometer (Edinburgh Instrument FLS 1000) with respective excitation source, 506 nm light irradiation from a Xenon arc lamp and 505 nm picosecond pulsed diode laser. Electroluminescence (EL) measurement was conducted by using a customized instrument from PL instrument. UV-Visible absorption measurement was conducted by a HITACHI UH4150 spectrometer. X-ray diffraction (XRD) measurement was conducted by an X-ray diffractometer (Malvern Panalytical Empyrean) with Cu Kα radiation. X-ray photoelectron spectroscopy (XPS) and Ultraviolet photoemission spectroscopy (UPS) spectra were acquired using an X-ray photoelectron spectrometer (PHI VersaProbe II Scanning XPS Microprobe). The XPS measurements utilized a monochromatic Al–Kα source with a photon energy of 1486.6 eV, while UPS measurements were conducted with a nonmonochromatic He–I source at an energy of 21.22 eV. The UPS measurements were undertaken at 5 V applied bias. The depth profile was carried out on the Time-of-Flight Secondary Ion Mass Spectrometry (ToF-SIMS, the Physical Electronics nanoTOF II ToF-SIMS instrument), with a 20 eV Ar gas cluster ion beam. Samples that are sensitive to air and moisture were introduced into a controlled environment within a glovebox and subsequently conveyed into the instrument using a vacuum transfer vessel. The particle size distribution spectra for perovskite precursor solutions were obtained using the dynamic light scattering technique (DLS, Zetasizer Nano ZS), with a light source of He-Ne 633 nm laser. The DLS measurements were undertaken at room temperature in the ambient environment.

As for device characterization, current density-voltage (J-V) characteristics was measured by a Keithley 2400 source meter (scan rate 10–50 mV/s) under simulated 100 mW/$cm^2$ AM 1.5 G light condition (calibrated by a reference silicon cell covered by KG5 filter glass, calibrated in 2020) produced by a 450 W Xenon lamp (Oriel Sol 2 A Class ABA). Black metal aperture masks were used during the J-V measurements for solar cells with dimensions calibrated by optical microscopy. Maximum power output tracking (MPPT) measurement was conducted under simulated 120 mW/$cm^2$ AM 1.5 G light condition produced by a solar simulator (Lumartix Solixon A-20) and measured by a Keithley 2400 source meter controlled by ReRa Tracer 3.1 software. The relative humidity and temperature were controlled by an RH meter and the silicone gel drier, and thermoelectric plates and a temperature meter with a cooling fan, respectively. The chamber was customized with well wiring and sealing. A thermal camera was also used to verify the surface temperature of different samples.

## Computation details

CASTEP calculation package was used for conducting the first principle calculation for the interaction between $MAPbI_3$ and biomolecules[47]. As for the simulation part in this work, it is important to employ the generalized gradient approximation (GGA) in the form of Perdew–Burker–Ernzehof (PBE) exchange-correlation function and set 517 eV as the cut-off energy for plane-wave basis. The Broyden–Fletcher–Goldfarb–Shanno (BFGS) algorithm was employed to perform the structural optimization, along with $10^{-5}$ eV/atom and 0.03 eV/Å set to be the convergence threshold for force and energy. In the electronic property calculation, the Brillouin zone of different structures was sampled by adapting the gamma-centered Monkhorst–Pack scheme and its $2 \times 2 \times 2$ k-point set.

## Reporting Summary

Further information on research design is available in the Nature Portfolio Reporting Summary linked to this article.

# Data availability

The data that support the findings of this study are provided in the main article and the Supplementary Information and are available from the lead corresponding author on request. Source data are provided with this paper.

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

## Acknowledgements

This material is based upon work supported by the U.S. Department of Energy's Office of Energy Efficiency and Renewable Energy (EERE) under the Solar Energy Technologies Office Award Number DE-EE0009364. J.Y. acknowledges the support from the U.S. Department of Energy Award No. DE-SC0019844, under the STTR program (Prime – NanoSonic

Inc.). A.K. acknowledges the support through the U.S. Department of Agriculture-National Institute of Food and Agriculture USDA-NIFA), under Award No. 2019-67021-28991. L.Z. acknowledges the support through Huck Institute of the Life Sciences seed grant (HITS). H.W. acknowledges the support through the Air Force Office of Scientific Research (AFOSR award number FA9550-20-1-0157). Kai.W. would like to acknowledge Prof. Congcong Wu's help (Hubei University, China) on DFT discussion. H.W. and Y.H. acknowledge Dr. Bangzhi Liu's help (Penn State University) on SEM measurement. The authors would like to acknowledge the support of Material Characterization Lab, Nanofabrication Lab, Materials Research Institute at Penn State University. "This report was prepared as an account of work sponsored by an agency of the United States Government. Neither the United States Government nor any agency thereof, nor any of their employees, makes any warranty, express or implied, or assumes any legal liability or responsibility for the accuracy, completeness, or usefulness of any information, apparatus, product, or process disclosed, or represents that its use would not infringe privately owned rights. Reference herein to any specific commercial product, process, or service by trade name, trademark, manufacturer, or otherwise does not necessarily constitute or imply its endorsement, recommendation, or favoring by the United States Government or any agency thereof. The views and opinions of authors expressed herein do not necessarily state or reflect those of the United States Government or any agency thereof."

## Author contributions

S.P. conceives the idea of bio-photovoltaics. S.P. and Kai.W. supervised and administrated the project. Kai.W. and H.W. designed the down selection concept. H.W. and Y.H. performed the material synthesis and characterization analysis, H.W., Y.H. and J.Y. designed and fabricated the solar modules, H.W., L.Z. and D.Y. performed the performance evaluation and related tests. Ke.W. performed the high-resolution TEM measurement and analysis. Y.H., J.Q. and Kai.W. performed the DFT calculation and analysis. A.K. and J.Q. contributed in-depth discussion to the whole work. Kai.W. and H.W. prepared the first version of the manuscript. All the authors have discussed the research data, reviewed the manuscript, and provided comments. Please note that authors D.Y. and Kai.W. were faculty at Penn State University when the work was completed. During the manuscript preparation stage, D.Y. and Kai.W. took new positions at other affiliations as indicated in the author information.

## Competing interests

The authors declare no competing interests.
