## [Peer Review File · Nature Communications]

Down-Selection of Biomolecules to Assemble “Reverse Micelle” with PerovskitesREVIEWER COMMENTS

Reviewer #1 (Remarks to the Author):

In this article, the authors used a series of types of biomolecules to regulate the morphology of perovskite films, and proposed a “reverse micelle” method via forming top nanoparticle layer onto the MAPbI₃ films. In addition, from Tier 1 to Tier 3, and from the small area cells and the large area modules, the author used many biomolecules to improve the device performance by DFT calculation and experiments. The manuscript of this work is supposed to have prominent significance to the photovoltaic community. The following questions should be carefully solved in major revision.

1. The artemisinin (ART) biomolecule has been reported to fabricate the “perovskite/perovskite” bilayer by refer. 21 (Nano Energy 78 (2020) 105133). Thus, the article needs to demonstrate the difference between the previous work and this work.

2. In Supplementary Fig. 9, different biomolecules have different molecular weights, and mol/mL should be more accurate for concentration optimization, and is not mg/mL.

3. Why does β -estradiol (hydroxyl) improve efficiency but reduce stability? Need to supplement relevant characterization explanations

4. As the carbonyl-based molecules, whether natural or not, have been widely used as additives to passivate and stabilized halide perovskites. The core and creative point in this work is the design of the “reverse micelle” in perovskites and subsequent bi-layer. Therefore, the universality is very important to prove the effects of reverse micelle on perovskites. Indeed, the authors presented an example in the mixed-cation CsFAMA system, but showing lower efficiency than MAPbI₃ devices and even inapplicable in FA-based or rich-FA based perovskites, can the authors give more detailed reasons?

5. The fabrication of large-area modules should use large-scale preparation methods, such as blade coating, rather than spin coating, which is not the fabrication method of Up-scalability. The related references is and

6. Why does the “reverse micelle” layer only form on the upper surface of the perovskite film, not at the buried interface? Whether the “reverse micelle” layer will penetrate into the mesoporous TiO₂, so it cannot be seen intuitively from the SEM images.

7. Why is camphor the best? Is it related to its molecular configuration or the thickness of the formed reverse micelle layer?

8. Regarding the problem of surface hydrophobicity, from the cross-section SEM images, the thickest of film based musone biomolecule has the largest water contact angle, which is reasonable, but the musone thick film is thinned, and whether FF can be improved to achieve high efficiency, because the corresponding Voc can reach 1.21 V. Camphor-based device has the best PCE, whether it is appropriate as target simple. SEM images of the perovskite films by different concentrations should be supplied and

study the relationship between the thickness and performance of the top nanoparticles layer, and the actual role of biomolecules should be carefully considered.

9. The length in Results section especially concerning some background introduction and basic interpretations those have been widely known in this field, such as the effects on Voc and corresponding formula derivation of recombination losses, should be shortened.

10. There is a special hysteresis phenomenon in perovskite solar cells, which can lead to deviations in true efficiency. Thus, the authors should provide the J-V curves of perovskite solar cells under both forward and reverse scanning, and analyse the hysteresis.

11. Regarding the interfacial energy band arrangement between perovskite and Spiro-OMeTAD, further experimental evidences such as UPS tests should be provided in addition to the work functions measured by KPFM.

12. The authors performed a series of stability tests for perovskite solar cells in different conditions, demonstrating higher stability in bio-molecules modified devices compared to those in pristine ones. As is well known, the stability is determined by not only the perovskite materials but also the charge transport layers. The Li-doped Spiro-OMeTAD generally plays a key factor in the light-heat and/or moisture instability of perovskite devices due to the the lower glass-transition-temperature as well as the fast diffusion of lithium ions and the hydrophilicity. Why this work showed such high stability even in harsh conditions (e.g., 1.2 sun, 85% RH, and 85°C)?

13. The PCE is improved mainly by the increased Voc, why is open circuit voltage improved so much? The authors attributed the reason to the less recombination losses induced by matched energy band arrangement. Any other reasons? More detailed characterization and analysis should be provided

14. The authors claimed they obtained the highest PCE among MAPbI₃-based devices reported so far. As this value has not been certified, I recommend they tune down the statement.

15. These 12, 20, 23, 24, 27, 31 references' journal abbreviations need correction.

Reviewer #2 (Remarks to the Author):

In this article, the authors select molecules inspired from biological macromolecules and refine their selection through the prediction of solar cell performance parameters. This is a large computational study conducted for a range of biomolecules and their derivatives. The screening process proposed by the authors is interesting for narrowing down the search of additives in perovskite-based solar cells.

The authors propose a reverse micelle formation with the biomolecules and the perovskite crystals. They then apply the materials to the fabrication of photovoltaic devices and characterize the devices.

This article is strong in terms of computational studies, and the photovoltaic device characterization is thorough, but I find that it lacks some experimental evidence to support the proposed molecular additive-perovskite interactions. Nonetheless, the screening performed and the improvement shown

through the three tiers of refinement demonstrate the validity of the approach. The degradation study comparing the hep-modified perovskite and the control is also convincing to demonstrate stabilization effect of the hep molecule.

Please see a few concerns and questions below:

- In some places in the article, I find that the word “biomolecules” has not been used appropriately. At the beginning of the introduction, the authors state “Biomolecules are organic compounds with several types of functional groups such as hydroxyl, methyl, carbonyl, carboxyl, amino, and phosphate groups,¹ playing important role in the formation of molecules like DNA, proteins, carbohydrates, and lipids.”, but this is misleading as this definition of biomolecules appears not to include molecules like DNA, proteins, carbohydrates and lipids, but only the smaller, more primitive molecules. Authors should clearly define which type of small molecules they have used in their studies. In addition, later on, authors mention the “functionalization of hydrocarbons” that can lead to changes in reactivity and interactions – thus, they refer to modified synthetic small molecules here, and not biomolecules. These molecules may be inspired by the structure of biomolecules, but they are not biological. The “Tier 1” molecules that are reported are biological, but the “Tier 3” molecules are not. I request that authors revise their terminology and clarify which types of molecules they used in their studies.

- Authors refer to the synthesis of perovskites with “molecular additives”, but they have not provided references for previous examples reporting the incorporation of biomolecules with perovskites. They do mention that “certain biomolecules have been reported in prior works”, but no references are provided. This is an entire body of literature that is lacking from the manuscript. For instance: Small, 13, 22, 1604305, 2017; Adv Func Mater 30, 42, 2005136, 2020 ; ACS Appl. Nano Mater. 2022, 5, 9, 12666–12678; and more.

- While the others have performed theoretical density of state calculations to determine interactions between the molecules tested and the perovskite lattice, I would have liked to see some experimental evidence of these interactions, for some of the molecules. Several material characterization techniques (spectroscopic techniques in particular) could have been used to validate interactions and to characterize the effects of the molecules of the optoelectronic properties of the perovskites.

- From the main manuscript, it is unclear how the authors have arrived at the conclusion that a reverse micelle was forming. Is there experimental evidence of the formation of these structures?

- The authors state that after 5mg/mL, the additives had negative effects on device efficiency and postulate that it is due to their insulating nature. Have they investigated what happens more specifically? Above a certain concentration, do the molecules still interact in the same manner with the perovskite crystals? Do they form structures separate for the perovskite? Do they accumulate in sections of the films?

- The text should be reviewed carefully for typos and a few grammatical mistakes. For instance, from the first paragraph only of the results section: “the halide perovskite is A metastable ...” “the mergence of point defectS, ...” “material degradation particularLY” “In natural systemS,...” ...

Reviewer #3 (Remarks to the Author):

In this manuscript, three tiers of biomolecules have been systematically tested and the results reveal that “reverse-micelles” consisting of biomolecules and precursors are involved in the perovskite crystallization. As a result, the core@shell perovskite@biomolecule nanoparticles are accumulated on the top of the perovskite film, leading to Fermi level splitting within the perovskite film and suppressed reverse charge transfer. By this reverse micelle strategy, excellent device performance could be achieved for either small-area or large-area solar cells or modules. This work deserves publication in nature communications. Nevertheless, several issues need to be clarified before acceptance.

1. The introduction focuses on the interaction between metallic compounds and bio-molecules in nature, which is too general to get the real point. The introduction shall focus on the recent advances of various perovskite solar cells with bio-molecules and point out the problems or challenges in this field.
2. The authors mentioned the motivation of this work in the results section. That is, “there lacks a systematic understanding on how a molecule can induce various effects at different length scales, as well as how different”. I suggest providing the motivation in the introduction section for better readability.
3. The new point in this manuscript is to establish the mechanism on how the “reverse-micelle” affects perovskite crystallization and its performance. The authors shall provide a more clear definition of the term “reverse micelle” in this manuscript and identify the size distribution of reverse micelles based on different bio-molecules.
4. The authors added acetonitrile into the ethanol solution of perovskite precursors. Please explain why acetonitrile and ethanol are used here. In fact, DMSO, DMF, and butyrolactone are more common solvents for perovskite precursors.
5. In Figure 1a, ball-stick models are provided for bio-molecules, but the problem is poor readability. Chemdraw structure would be better here for higher clarity.
6. In Figure 1a, DNA was classified into the class with PO4 group. Nevertheless, there are also amino and amide groups in DNA. In NAM, there is not only amino group but also -CO- group. So, why DNA or NAM is classified into the class with PO4 or amino? Please explain why PO4 or amino group has superiority when considering its classification.
7. Fig. 1c shows the film drying process where the “reverse micelle” structures are firstly assembled in the solution state, accumulating at the top surface of the wet membrane. However, there lacks direct evidence to support this proposal. In Figure 3d, the HRTEM image reveals there is an amorphous layer on the surface of the perovskite. To verify this amorphous layer being from camphor, ToF-SIMS and XPS analysis shall be provided. To verify the formation of reverse micelle in the precursor solution, light dynamic scattering analysis shall be provided. Is it possible that the reverse micelles also accumulate at the buried interface of perovskite film (the bottom side of perovskite film)? The authors shall consider or exclude such a possibility by depth-dependent analysis (i.e., ToF-SIMS).

8. As illustrated in Figure 1a, the reverse micelle is formed involving both bio-molecular and perovskite precursors, which is reasonable. So, again I suggest that light dynamic scattering analysis shall be performed to identify the size distribution of reverse micelle in precursor solution.
9. The authors claim that there is Fermi level splitting within the perovskite film, which will enhance the built-in electric field in the regular structured device. To confirm this conclusion, disproof shall be provided. For example, when the camphor doped perovskite film is applied to inverted device, the Fermi level splitting will weaken the built-in electric field and lower the device performance.
10. In Figure 3a, the camphor modified devices deliver the largest VOC, while its PCE is not the highest. Please discuss how the various reverse micelles affect the FF and JSC to varying degrees.
11. It is hard to read the letters in Figure 5c. Please modify it.
12. In page 13, the correct sentence is as follows: Fig. 4c(i) shows the PL results of samples consisting of perovskite with a top Spiro-OMeTAD HTL.
13. Why the perovskite@bio-molecule nanoparticles have a larger bandgap.

Reviewer comments are highlighted in brown.

Our responses are in black. (References used in the response are listed in the end of this response letter)
Additional or revised sentences cited from the revised manuscript or Supplementary Materials are highlighted in blue.

Reviewer #1 (Remarks to the Author):

In this article, the authors used a series of types of biomolecules to regulate the morphology of perovskite films, and proposed a “reverse micelle” method via forming top nanoparticle layer onto the MAPbI₃ films. In addition, from Tier 1 to Tier 3, and from the small area cells and the large area modules, the author used many biomolecules to improve the device performance by DFT calculation and experiments. The manuscript of this work is supposed to have prominent significance to the photovoltaic community. The following questions should be carefully solved in major revision.

Responses: We sincerely appreciate your time and patience spent in evaluating our manuscript, and we extend our thanks for your thoughtful evaluation that “The manuscript of this work is supposed to have prominent significance to the photovoltaic community” as well as your constructive comments, which have undeniably contributed to the enhancement of this work. In light of your comments, we have meticulously addressed each concern in our revised version, by adding supplementary experiments and discussions to support our findings. We believe that these enhancements have solidified the basis of this work.

1. The artemisinin (ART) biomolecule has been reported to fabricate the “perovskite/perovskite” bilayer by refer. 21 (*Nano Energy* 78 (2020) 105133). Thus, the article needs to demonstrate the difference between the previous work and this work.

Responses: Thank you for this insightful comment. We understand that the use of the artemisinin (ART) biomolecule in fabricating the “perovskite/perovskite” bilayer has been reported in reference 21 (*Nano Energy* 2020, 78, 105133). However, the detailed mechanistic understanding of ART biomolecule’s contribution to the performance enhancement remains elusive due to its complex chemical structure. In addition, replication of the efficacy of this strategy will need further verification of similar molecules to ART, which remains unexplored in the prior work.

The selection of a suitable molecular additive for perovskite solar cells is a multi-faceted process involving numerous factors spanning from the molecular to the device level. Most previous works focused on one particular aspect such as defect passivation at the lattice level, leaving the overall understanding fragmented. Our study aims to bridge this gap by providing a more comprehensive perspective, correlating multiple factors to device performance.

In this work, a comprehensive library of 16 different molecules, with various concentrations, has been investigated. Among these, only five molecules have been reported previously with an emphasis on one particular beneficial aspect of their use. However, we moved beyond this single-molecule focus by also including 11 unreported molecules.

Table R1 below summarizes our systematic approach, noting the molecules used, their suspected mechanism of action, and previous references, if applicable. Notably, the molecules listed in Tier 2 and Tier 3 are newly introduced to this field in our study, with Hexane used as a reference.

Table R1 Molecules used in this work showing report history.

Molecule	Mechanism	Format Reference
Tier 1:		
Niacin	Defect passivation	J. Mater. Chem. C, 2021, 9, 6217-6224
Artemisinin	Band engineering	Nano Energy 78 (2020) 105133
DNA	Hole transfer optimization	ACS Energy Lett. 2019, 4, 2646–2655
Estradiol	This work	n.a.
Nicotinamide	Surface and grain boundary passivation	ACS Appl. Mater. Interfaces 2020, 12, 47, 52500–52508
Tier 2:		
Citral	This work	n.a.

Cinnamaldehyde	This work	n.a.
Camphor	This work	n.a.
Menthol	This work	n.a.
Muscone	This work	n.a.
Tier 3:		
Hexane	n-Hexane surface treatment to eliminate DMF residual (not introduced inside of perovskite like this work)	J. Mater. Chem. A, 2015, 3, 22839-22845
Glutaraldehyde	This work	n.a.
Heptanal	This work	n.a.
Cyclohexanal	This work	n.a.
Enanthone	This work	n.a.
Tricosanone	This work	n.a.

Here we would also like to re-introduce the logic of this work.

(1) The background knowledge suggests many biomaterials have been used in perovskite solar cells to induce positive effects on device performance by various mechanisms (**Table R2**) and various perspectives (lifetime, thermal-/moisture-/operational-/shelf-stability, efficiency (either J_{sc} , or V_{oc} or FF), etc.), but there is no universal agreement on clarifying the logic chain from molecular level, to meso-level, to interface level, to device level, and to module level, globally.

For example, **Fig. R1** below shows this background information. Top panel lists examples of prior works, where mostly have a specific focus on either defect passivation, or crystallization modification, or others, but *oversighting other possible effects in the meantime, for example, how the additive can affect at different length scales*. While in real case (**Fig. R1** bottom panel), many effects occur at different length scales can be either synergistically or contradictorily affect the overall device performance. Furthermore, the device performance manifested in different scenarios (either lifetime, or efficiency, or other figure-of-merits such as hysteresis, functionalities of self-healing etc.) can also be affected significantly different between each other. In another word, the ‘hidden Blackbox’ linking the methods (e.g., introduction of a bio-additive) to the results (i.e., device performance) can be an “interpenetrating network” consisting of various mechanisms at different length scales. Therefore, simply focusing on a single perspective (e.g., passivation at molecular level) without jointly considering grain-, interface-, device-level effect, can be less convincing to explain the real device performance in practice. There will need to be a comprehensive understanding of how the additive triggers what exact effects at each length-scale, and how these different effects can lead to different changes in all figure-of-merits of device performance. Contrast to those prior researches, this work provides these aforementioned points, providing the first exploration on how different biomolecules systematically affect these aspects at various length scales.

Table R2 Other molecules from prior reports showing various mechanisms.

Biomolecule	Mechanism	Format Reference
capsaicin	p-n junction	Joule 5, 467–480, February 17, 2021
caffeine	cross-link	Joule 3, 1464–1477, June 19, 2019 1465
trimesic acid	trap-state passivation	J. Phys. Chem. C 2019, 123, 14223–14228
ethyl cellulose	cross-anchoring	ACS Appl. Mater. Interfaces 2019, 11, 14, 13491–13498
2-pyridylthiourea	crystal engineering, hydrogen bond formation	J. Mater. Chem. A 2017, 5, 13448.
1,3,4-thiadiazolidine-2,5-dithione	crystal engineering, uniformity improvement	J. Mater. Chem. A, 2018,6, 4971-4980
1,3-diaminopropane	trap-state passivation	Sci. Adv. 2019; 5 : eaav8925
monoammonium zinc porphyrin	crystal engineering and trap-state passivation	J. Am. Chem. Soc. 2019, 141, 6345–6351
2-aminoethylphosphonic acid	trap-state passivation	J. Am. Chem. Soc. 2020, 142, 20071–20079

urea	crystal engineering, grain boundary passivation	Chem 3, 290–302, August 10, 2017
thiourea	Lewis adducts formation, passivation	Sol. RRL2018,2, 1800034
maize starch	crystal engineering, template function	ACS Appl. Energy Mater. 2021, 4, 11194–11203

In fact, different mechanisms can trigger a series effect throughout length-scales from molecular level to meso- and interface-levels

- There lacks a comprehensive vision: how a bio-additive can trigger the what chain effect over different length scales?
- There lacks an in-depth understanding: how the nominal performance enhancement is correlated to what fundamental science?
- Discovery of new colloidal manifestation in crystalline system – ‘reverse micelle’ crystal
- Down-selection material screening leads to a ‘general rule of selection’

Fig. R1. Background information. Top panel: examples of prior works using additives and mostly introduce one specific mechanism to explain the device improvement. However, in real case, many effects at different length scale (bottom panel) can interactionally lead to different influence on device. Furthermore, their individual, synergistic, and contradictory effects on different device features of stability, efficiency, and other factors can also vary from one to one. Overall, device performance can be very complexly affected from an additive. Elucidating this ‘hidden Blackbox’ linking the additive and the device results will not only need comprehensive understanding of one molecular effect over the whole length scale but also the verification of different molecules of the same type. That is, matrix-type research, which is exemplified by this work.

(2) Logic:

In **Tier 1** class, we selected five representative biomolecules meeting requirements of (i) priorly reported (to expand from prior single focus towards radar plots on how the molecule can affect other aspects of the device) and (ii) have the characteristic chemical groups representing the main five chemical types in natural biomolecules. After a joint consideration on efficiency (broken down into FF, J_{sc} , V_{oc}), lifetime, cost, etc.,

we screen towards one group with highest promise, this goes to the **Tier 2** class. The **Tier 2** class molecules have similar functional groups to the chosen one in **Tier 1**, but it is further broken down into two types: aldehyde and ketone. We evaluate the **Tier 2** class at (i) small cell device level, in terms of efficiency, hydrophobicity, and discovered (ii) the “reverse micelle” structure with in-depth understanding of how this spontaneous design can affect at different length scales (fundamental investigation). From these results, a rough rule-of-selection assumption is conceptualized. We then use **Tier 3** class molecules to (i) verify this assumption (proof-of-concept) and (ii) validate it at module level. Various stability testing converges to the conclusion that this “reverse micelle” can significantly improve the device performance.

We appreciate this point, and we have added the following content in the main text:

“In summary, our work seeks to offer a comprehensive understanding of the influence of various molecular additives on perovskite solar cell performance from molecular scale to module level, providing a useful guide for future research in this field.”

2. In Supplementary Fig. 9, different biomolecules have different molecular weights, and mol/mL should be more accurate for concentration optimization, and is not mg/mL.

Responses: We appreciate this thoughtful comment. In general case, we agree that the molar concentration is supposed to be more accurate to mass concentration. While in specific cases such as the comparison between NAM and DNA, there exists a significant disparity in molecular weights (NAM: 123.11 mg/mmol vs. DNA: ~2000 bp (1 bp = 660 g/mol), ~1.32×10⁶ mg/mmol). In these cases, a direct comparison based on molar concentration may not yield rational meaning. Additionally, DNA’s structure, being a double-helix chain comprised of variable A-T and C-G pairs, complicates the calculation of an average molar mass of its repeating units.

In light of these complexities, we opted to use mass concentration (mg/mL) as a simpler, more practical index. Given that the biomolecules we investigated primarily contain Carbon (C), Oxygen (O), and Nitrogen (N), and considering their similar relative molar masses (C:12, O:16, N:14), using mass concentration allows us to roughly estimate the information of atomic numbers. This estimation is based on the idea that since the atomic masses of these elements are somewhat similar (C:12, O:16, N:14), a higher mass concentration would typically imply a greater number of atoms. In the context of these specific biomolecules, using mass concentration can provide an indicative estimate of the number of atoms present.

3. Why does β-estradiol (hydroxyl) improve efficiency but reduce stability? Need to supplement relevant characterization explanations

Responses: We appreciate this comment. The β-estradiol (Esd) molecule has two hydroxyl groups, which can have molecular interaction with perovskite materials to passivate the defects, according to previous research (*ACS Energy Lett.* 2020, 5, 10, 3268–3275). Here, we are trying to conduct the PL and TRPL measurement to verify the defect-passivation effect of β-estradiol molecule. In **Fig. R2a** (shown below), compared to pristine perovskite film on the glass substrate, the one with β-estradiol shows an enhanced photoluminescence intensity, which indicates the radioactive recombination is improved or the nonradioactive recombination is suppressed. Besides, the peak position was blue-shifted from 768.5 nm to 765.5 nm after the corporation of β-estradiol molecule due to the interaction between perovskite and biomolecules, leading to the decrease of trap density (*Nat. Commun.* 2014, 5, 5784). The TRPL test was conducted to investigate the kinetics of charge carriers such as charge carrier lifetime, as shown in **Fig. R2d**, and the spectra can be fitted by the biexponential decay function (*Adv. Mater.* 2018, 30(35), 1801418):

$$f(t) = A_1 \exp\left(\frac{-t}{\tau_1}\right) + A_2 \exp\left(\frac{-t}{\tau_2}\right) + B$$

In which A_1 and A_2 refer to the decay amplitude, τ_1 and τ_2 refer to the fast and slow decay lifetime, B is the constant. The fast decay lifetime is regarding to the surface traps, the slow decay lifetime is regarding to the bulk traps (*Nat. Commun.* 2016, 7, 10214). With the help of biomolecules, the fast decay lifetime is elongated from 0.93 ns to 1.78 ns, which is nearly doubled. The stability of the corresponding perovskite films with or without biomolecules was also studied by PL and TRPL measurement at 2 h and 5 h, as shown in **Fig. R2b, c, e, f**, the Esd-perovskite film shows a faster PL intensity quenching compared to the pristine

one. By combining the fast and slow decay lifetime, the average lifetime can be obtained according to the following equation (*Adv. Energy Mater.* 2018, 8(3), 1701757):

$$\tau_{ave} = \frac{\sum A_i \tau_i^2}{\sum A_i \tau_i}$$

The average decay lifetime can be used as an index to indicate the stability of perovskite thin films, for the pristine one, the average decay lifetime at 0 h, 2 h and 5 h, changing from 5.93 ns, 5.56 ns, to 4.28 ns, however, for the Esd-perovskite one, the number varies from 5.46 ns, 4.71 ns, to 3.14 ns. The contact angle measurement in **Fig. R2g&h** shows that the Esd-perovskite film has worse hydrophobicity with a smaller contact angle of 26° compared with 29° for the pristine film. These results indicate that the Esd-perovskite has a better performance but decreased stability.

Fig. R2. (a) PL spectra of pristine perovskite film and Esd-perovskite film. (b, c) PL spectra of pristine perovskite film and Esd-perovskite film at 0 h, 2 h, 5 h. (d) TRPL spectra of pristine perovskite film and Esd-perovskite film. (e, f) TRPL spectra of pristine perovskite film and Esd-perovskite film at 0 h, 2 h, 5 h. (g, h) Water contact angle measurement of pristine perovskite film and Esd-perovskite film.

4. As the carbonyl-based molecules, whether natural or not, have been widely used as additives to passivate and stabilized halide perovskites. The core and creative point in this work is the design of the “reverse micelle” in perovskites and subsequent bi-layer. Therefore, the universality is very important to prove the effects of reverse micelle on perovskites. Indeed, the authors presented an example in the mixed-cation CsFAMA system, but showing lower efficiency than MAPbI₃ devices and even inapplicable in FA-based or rich-FA based perovskites, can the authors give more detailed reasons?

Responses: We thank reviewer for this comment and agree well that other compositions such as Cs-doped FAMA-based perovskite can be more promising for practical use. These Cs-doped FAMA-based perovskite has a main component of FA where the mixed A-site cations ingredient of Cs and MA is used to release the strain and stabilize the phase (as FA is larger cation which can induce strong lattice distortion which will need smaller cations to compensate the stress). To make it simpler, we use “FA-based” perovskite for the following discussion.

Comparing “MA-based” and “FA-based” perovskite: Since the [PbI₃]⁻ sublattice is identical between them, the FA-based perovskites have more hydrogen bonds between FA⁺ and the [PbI₃]⁻ sublattice (shown below in **Fig. R3**), rendering higher stability. Meanwhile, the larger FA⁺ cation leads to a tolerance factor closer to 1, i.e., a more cubic-like unit cell, which gives better band structure responsible for higher V_{OC} and J_{SC} in solar cell devices compared to their MA-based counterparts. Back to molecular scale, it should be noted that there are ca. 20% more numbers of hydrogen bonds in FA-based perovskite compared to the MA-based perovskite.

Fig. R3 Hydrogen bonds between A-site cation with the $[BX_3]$ sublattice in case of FA and MA, (Image courtesy: computational materials physics group, Shuxia Tao).

Experiment with FA-based perovskite: When shifting from MA-based perovskite to FA-based perovskite, the extra hydrogen sites of FA^+ also induce a stronger solvation effect, which makes it more difficult to crystallize the FA-based perovskites in the colloidal system. It should be noted that: different from traditional DMSO/DMF system, the solution system used in this study consists of ACN/Et-OH/MMA in order to be inclusive with biomolecules to form the colloidal system. As we have discussed in the main text, crystallization and biomolecular colloidal chelation needs to be balanced to form the “reverse micelle”.

We have tried different stoichiometric ratios between FA/MA with Cs-dopant and different solution conditions to understand how the micelle concept can be applied to these systems. As discussed above, under high FA/MA ratio (e.g., $FA_{0.8}MA_{0.15}$ in **Fig. S27**), it is of challenge to form a black “perovskite” film and modulating the FA/MA ratio can lead to a better crystallization but the cross-sectional SEM reveals a distinctly different morphology from the “reverse micelle”. Insulative phase on top of the perovskite film and accumulated phases are observed in the sample but the micelle structure is missing, which is due to the imbalanced colloidal-crystallization-solvation kinetics in the precursor solution. Since the FA can be more easily solvated, we then intentionally add more nonpolar solvent (THF in this example) to balance the solvation of the biomolecules. As a result, the familiar bilayer perovskite with the top being “reverse micelle” crystals are observed in the final film. Thus, the compatibility of this strategy is verified to other perovskite compositions, which will conditionally require a polarity tuning for the precursor solution system.

Inferior device performance in the FA-based perovskite: We appreciate Reviewer #1 highlighting the seemingly lower PCE of the FA-based device compared to the MA-based one, which at first glance appears to contradict conventional understanding. Ideally, given the same crystalline characteristics, FA-based devices should outperform MA-based ones. However, as demonstrated in **Fig. S27a**, the bilayer FA-film showcases smaller crystalline grains in its bottom layer, and various defects are evident. This deviation can be attributed to the requirement of this “reverse micelle” colloidal solution developed in our study. Achieving optimal emulsification properties in the precursor solution requires meticulous tuning of all precursor components jointly, an aspect not fully addressed in this study. As a result, optimizing the polar modulation within the solution to improve crystalline features presents an opportunity for enhancement. We believe that this is the fundamental reason for the reduced performance observed in the FA-based perovskite device.

5. The fabrication of large-area modules should use large-scale preparation methods, such as blade coating, rather than spin coating, which is not the fabrication method of Up-scalability. The related references is and Responses: We appreciate this comment (it seems there is a missing part of this comment. We would be grateful if Reviewer #1 could provide further details or direct us to any relevant references. We will be more than happy to review and potentially incorporate them in our upcoming revision).

Technically, we agree well with that blade/slot-die coating could serve as more suitable large-scale preparation methods for upscaling consideration, compared to spin-coating. While spin-coating is effective for lab-scale applications and can offer excellent film uniformity, as illustrated in **Fig. R4**, its utility is primarily

for demonstrating concepts. In this research, which is fundamentally about proving a scientific concept, our priority was to leverage reliable, low-risk manufacturing techniques, leading us to use spin-coating for demonstrations.

Fig. R4 Cross-sectional SEM images of the MAPbI₃ film by spin coating method, captured at varying levels of magnification, demonstrate the film's uniform thickness and consistent crystalline structure across a substantial area.

6. Why does the "reverse micelle" layer only form on the upper surface of the perovskite film, not at the buried interface? Whether the "reverse micelle" layer will penetrate into the mesoporous TiO₂, so it cannot be seen intuitively from the SEM images.

Responses: We thank the reviewer for these great thoughts. The formation of the "reverse micelle" structure results from a self-assembly mechanism within our intricate precursor solution system. This system encompasses the ionic precursors of perovskite (e.g., MA⁺, Pb²⁺, I⁻), intermediate states stemming from the assembly of specific precursors (like the MA-Pb-I intermediate clusters), standalone biomolecules, and clusters that have adsorbed biomolecules (as depicted with a dashed cycle in **Fig. R5**).

These clusters boast a perovskite core enveloped by a biomolecule shell. The shell, owing to its organic nature and lightweight atomic constituents, reduces the cluster's overall mass density. Meanwhile, the perovskite in the same solution but devoid of these biomolecules exhibits a higher mass density. When the wet film forms, the clusters with a lighter mass tend to rise to the solution's surface, while the denser, pure perovskite gravitates toward the bottom. This phenomenon is reminiscent of an oil-water emulsion, where oil naturally rises. Consequently, the "reverse micelle" layer primarily develops on the film's upper surface, leaving the unaltered perovskite layer below.

Fig. R5 Hypothetical formation mechanism of the perovskite/perovskite bilayer structure where the top perovskite nanocrystals exhibit higher energy states whereby leading to a higher potential to the device.

7. Why is camphor the best? Is it related to its molecular configuration or the thickness of the formed reverse micelle layer?

Responses: We appreciate this comment. When examining **Fig. R6a** (shown below), which displays the cross-section SEM images of perovskite films fabricated with all the **Tier 2** biomolecules (all at a concentration of 3.0 mg/mL), we can discern the thickness of the top layer corresponding to each molecule. This is further quantified in **Fig. R6c**. Among these, the film developed with camphor exhibits the thinnest (lowest magnitude) and most uniform top layer (smallest deviation).

In parallel, as depicted in **Supplementary Fig. 10**, solar cells based on camphor-infused perovskite achieve the highest fill factor (FF), and consequently, the top efficiency. The FF in solar cell device is closely related to the charge carrier transport feature in device, the thinner and more uniform reverse micelle layer is expected to be less insulative and thus be responsible for a better transport behavior among other biomolecule-treated films. Consequently, the highest PCE was observed in the camphor-perovskite device.

The variance in the thickness of the reverse micelle layer, even when using biomolecules at the same concentration, can potentially be attributed to the differing strengths of polar-head attractions from the biomolecules to the perovskite core within the reverse micelle nanoparticle. Factors such as steric effects, interactions between the nonpolar tails of the biomolecules and solvent molecules, and the neighboring influences of adjacent biomolecules bound to the same perovskite core may collectively result in discrepancies in the size, density, shape, and volume of the reverse micelle particles from case to case.

Fig. R6 (a) Cross-section SEM images for perovskite layer incorporated with different Tier 2 biomolecules. (b) Statistical analysis on PCE from solar cells (20 devices for each group) using Tier 2 class molecules. (c) Thickness summary of top layers for corresponding perovskite layers.

8. Regarding the problem of surface hydrophobicity, from the cross-section SEM images, the thickest of film based musone biomolecule has the largest water contact angle, which is reasonable, but the musone thick film is thinned, and whether FF can be improved to achieve high efficiency, because the corresponding Voc can reach 1.21 V. Camphor-based device has the best PCE, whether it is appropriate as target sample. SEM images of the perovskite films by different concentrations should be supplied and study the relationship between the thickness and performance of the top nanoparticles layer, and the actual role of biomolecules should be carefully considered.

Responses: We thank Reviewer #1 for raising the discussion on the possibility of further improvement of device efficiency as well as your kind advice of additional experiments, which we think is thoughtful and constructive. As seen in the PCE-thickness dependance plot of **Tier 2** biomolecules (in above **Figs. R6c&d**), the thickness of the reverse micelle layer indeed influences the PCE. Specifically for each biomolecule, the reverse micelle thickness is also expected to have influence on the device performance. To verify this and following Reviewer #1's suggestion, we have conducted extra experiments using the two representative **Tier 2** molecules, i.e., muscone and camphor, under varying concentrations—from 0 mg/mL, to 1 mg/mL, 3 mg/mL, and 5 mg/mL. The SEM cross-sectional images, as presented in **Figs. R7a&b** (below), show a clear correlation between molecule concentration and the thickness of the top reverse micelle perovskite

layer. For both cases, peak PCE was observed at a concentration of 3 mg/mL. This suggests an “optimal thickness range” prevalent under such condition of a biomolecular concentration of ca. 3 mg/mL. For a closer look, **Fig. R7c** provides SEM images of camphor-based perovskite cells at these concentrations. A progressive increase in the thickness of the top reverse micelle layer is evident as concentration escalates. A parallel trend is observed in muscone-based cells, as shown in **Fig. R7d**. Detailly, at a concentration of 1 mg/mL, the top layer is not fully formed until the concentration increases to 3 mg/mL. While at 5 mg/mL, the quality of the bottom pristine perovskite layer decreases, characterized by smaller grains and a higher void density. In short, we agree well with Reviewer #1 that tuning the thickness of the reverse micelle layer can indeed modulate the device performance. While noting that the thickness is determined by the concentration of the biomolecules, the fluctuation of this concentration can also lead to synergistic effects such as the deviation of the crystalline feature of the bottom perovskite. The device performance is a holistic result integrated from all these effects at microscale, which are originated from the molecular interplay between the biomolecules and the colloidal solution.

Fig. R7 (a) J-V characteristics of perovskite solar cells with the addition of camphor molecules at different concentrations. (b) J-V characteristics of perovskite solar cells with the addition of muscone molecules at different concentrations. (c) Cross-sectional SEM images of camphor-based perovskite solar cells with

different doping concentrations. (d) Cross-sectional SEM images of muscone-based perovskite solar cells with different doping concentrations.

9. The length in Results section especially concerning some background introduction and basic interpretations those have been widely known in this field, such as the effects on V_{oc} and corresponding formula derivation of recombination losses, should be shortened.

Responses: We appreciate this comment. We have simplified the discussion of this part and moved the original detailed interpretations into the **Supplementary Note 3** in the revised **Supplemental Information** file.

To make it clear, the revised narratives of the corresponding discussion are cited here for your reading convenience:

“To verify the reduced losses of recombination, we further investigate the recombination process of different solar cell devices. The light current density-voltage (J-V) curve in **Fig. 4a(i)** highlights a noticeable improvement in V_{oc} when comparing the pristine device with the *camphor*-MAPbI₃ device. This improvement reflects a concurrent rise in V_{oc} (around 10%) and FF when incorporating the biomolecule. In essence, the FF reflects the charge transport feature in device, particularly predominant at smaller internal fields. **Fig. 4a(ii)** manifests the comparison in this perspective. By plotting photocurrent ($J_{ph} = J_i - J_d$) vs. effective bias ($V_{eff} = V_0 - V$) (where J_i is the current density under illumination, J_d is the dark current, V_0 is the voltage when $J_{ph} = 0$, and V is the applied bias voltage, respectively),³⁷ we observed higher J_{ph} in the *camphor*-MAPbI₃ device under lower V_{eff} region (< 0.15 V), suggesting a more efficient charge transport behavior in such device (detailed explanation is incorporated in **Supplementary Note 3**). In addition, light-dependent investigations have been conducted for both devices under both short- and open-circuit conditions. As seen in the light-dependent V_{oc} plot in **Fig. 4a(iii)**, the less deviated slope of 1.72 kT/e from *camphor*-MAPbI₃ device than that of 1.95 kT/e from the pristine device with regard to the ideal case (1 kT/e), suggests reduced trap-assisted recombination⁴² (Shockley-Read-Hall, SRH recombination) in the *camphor*-MAPbI₃ device. Consistently, **Fig. 4a(iv)** shows the power-law fitting of the log-log plot of J_{sc} vs. light intensity, smaller deviation of the power index in the *camphor*-MAPbI₃ device ($\alpha = 0.94$) than that of pristine device ($\alpha = 0.88$) with perspective to the ideal case ($\alpha = 1.00$), reveals a minimized recombination losses (detailed explanation is incorporated in **Supplementary Note 3**).

In addition to the electric results discussed above, a deeper understanding of the photophysical process is equally important. It should be noted that simply utilizing PL study to reveal the charge extraction efficiency is unrigorous. As seen in **Fig. 4b(i)**, photoexcitation can lead to Shockley-Read-Hall (SRH) recombination losses, PL emission, and extraction by HTL. Observation of PL intensity change remains inadequate to conclude the ratio between SRH losses and extraction by HTL, where the former is harmful to device performance but the latter is beneficial. In real device working conditions, due to the presence of internal field (asymmetric electrodes), the charge transfer across the perovskite/HTL interface is dominated by a drifting process rather than a simple diffusion, which will make the above model less applicable for analyzing these real cases. Thus, additional information is needed to identify the respective contributions from both SRH losses and extraction. Here we introduce the supplementary investigation by electroluminescence (EL) using the solar cell device but reverse the charge flow by current injection. As schematized in **Fig. 4b(ii)**, in condition of an identical current injection, the injected electrons can have three pathways of SRH recombination losses, electroluminescence

(bimolecular recombination) emission, and losses during charge transfer (CT) across multiple interfaces and within layers. Both losses to SRH recombination and CT are negatively contributing to the device. Thus, compared to PL, the result from EL can be a more direct index to evaluate the device performance. We carry out both PL and EL investigations for devices using either pristine or *camphor*-MAPbI₃ perovskite. **Fig. 4c(i)** shows the PL results of samples consisting of perovskite with a top Spiro-OMeTAD HTL. The pristine sample displays a spectrum with a peak around 1.58 eV which is consistent to the bandgap of MAPbI₃. In contrast, the *camphor*-MAPbI₃ perovskite displays a blue shift (**Supplementary Fig. 18b(ii)**), exhibiting a peak at 1.63 eV, accompanied by lower states peaked at 1.58 eV. The larger bandgap of 1.63 eV is most likely belong to the particle-like perovskites in the upper layer (**Supplementary Fig. 12c**) while the lower states (1.58 eV) are consistent to the bottom columnar grains identical to the pristine. As mentioned above, albeit the *camphor*-MAPbI₃ perovskite displays lower PL intensity, it is difficult to distinguish the contribution from either SRH recombination or charge extraction. Similarly, transient PL (**Supplementary Fig. 18b(iii)**) with information of photocarrier lifetime also cannot distinguish either. While the EL results in **Fig. 4c(ii)** reveal a clearly conclusive result. The *camphor*-MAPbI₃ device exhibits a 5-fold higher EL intensity than the pristine one, suggesting reduced losses through SRH or CT or both, which is consistent with the results of recombination studies. Similarly, two sub-peaks were observed which are centered at 1.58 and 1.63 eV, respectively, from the *camphor*-MAPbI₃ device (**Supplementary Fig. 18c(iv)**). This is consistent to the PL results and suggests that besides the conduction band frontier states (corresponding to 1.58 eV), there is an additional excited state of electrons at an energy level 50 meV higher (corresponding to 1.63 eV) (**Supplementary Fig. 18c(v)**). The higher energy state can also contribute to a higher potential energy to the excited electrons, jointly with the reduced recombination losses, leading to the higher V_{oc} of the devices with **Tier 2** class biomolecules.”

10. There is a special hysteresis phenomenon in perovskite solar cells, which can lead to deviations in true efficiency. Thus, the authors should provide the J-V curves of perovskite solar cells under both forward and reverse scanning, and analyze the hysteresis.

Responses: We acknowledge the significance of the comment provided. In our revised version, we have incorporated diligent steps to investigate the hysteresis behavior of our module devices. We employed a variety of conditions, utilizing nonpolar Hexane (Hex) and bipolar Glucose (Glu) which do not result in micelle formation, as well as amphiphilic biomolecules such as Heparin (Hep), Cyclodextrin (Cyc), Dihydrolipoic Acid (Dih), and Lauric Acid (Lau) that lead to the formation of a “reverse micelle” (r-micelle) structure. To provide a detailed analysis, we have compared the hysteresis indexes (HI), calculated using the formula ($HI = \frac{PCE_{reverse\ scan} - PCE_{forward\ scan}}{PCE_{reverse\ scan}}$) along with the PCEs obtained from different scanning directions. These comparisons are updated in the modified **Supplementary Fig. 21F** provided in the **Supplementary Information**.

From our observations, it is evident that the biomolecules capable of inducing the r-micelle structure (Hep, Cyc, Dih, Lau) demonstrate a lower HI in comparison to the reference group. On the other hand, Hex and Glu, which do not result in r-micelle formation, exhibit HIs that are similar to or even higher than that of the reference group. **Supplementary Fig. 21f** offers a thorough exploration of hysteresis across various device configurations. We scrutinized both small area and module devices, particularly highlighting the Hep-devices for their ability to show reduced hysteresis in both (i) small area cell devices and (ii) mini-module devices. In part (iii) of the same figure, we present a module-level comparison of PCEs obtained from different scan directions alongside the hysteresis index. Consistently, the biomolecules associated with the “reverse micelle” formation (Hep, Cyc, Dih, Lau) showcase a lower HI compared to the reference, while Hex and Glu exhibit higher HIs. We attribute the diminished hysteresis in “reverse micelle” devices to the terminal atomic anchoring effect exerted by the ligands. This effect plays a crucial role in minimizing the ionic contributions within the perovskite structure, thereby stabilizing the device performance across

different scanning conditions. Ionic motion is believed to be one of major origins of hysteresis in perovskite solar cell. The ionic motion can either be present within the grain or at the GB. **Fig. R10** has shown that at the GB there are various ionic species which can contribute to the ionic motion in the pristine perovskite conditions. In contrast, the wrapping of biomolecules can sufficiently reduce these ionic species, at least anchoring the terminal partially bonded atoms at the lattice edge. Therefore, we believe this ionic suppression can be the main driving force to the lower HI.

Adapted Supplementary Fig. 21f: Hysteresis investigation. We checked both the small area and module devices: the Hep-devices show smaller hysteresis in both (i) small area cell devices and (ii) mini-module devices. (iii) Module level comparison on PCEs scanned at different directions and hysteresis index, calculated from equation of $HI = \frac{PCE_{reverse\ scan} - PCE_{forward\ scan}}{PCE_{reverse\ scan}}$ (S10),⁵⁶ where $PCE_{reverse\ scan}$ is the module PCEs obtained from reverse scan and $PCE_{forward\ scan}$ is the same module PCEs obtained from forward scan. All the “reverse micelle” related biomolecules (Hep, Cyc, Dih, Lau) exhibit lower HI compared to reference, whereas the nonpolar Hex and bipolar Glu show higher HI. The reduced HI in “reverse micelle” device can be ascribed to the terminal atomic anchoring effect by the ligand which minimizes the ionic contribution in the perovskite.

Fig. R10 Transport analysis between regular crystal grain boundary and the “reverse micelle” crystal boundary.

11. Regarding the interfacial energy band arrangement between perovskite and Spiro-OMeTAD, further experimental evidences such as UPS tests should be provided in addition to the work functions measured by KPFM.

Responses: We appreciate this comment. We have conducted the UPS measurements on the pristine perovskite film and camphor-based film, as shown in **Fig. R11** below, the Fermi level can be obtained according to the calculation of (based on literature of (*Nat. Commun.* 2020, 11, 1245)):

$$Fermi\ Level = UV\ He(I) - Onset - Bias$$

In this case, UV He (I) is 21.22 eV, bias is 5 V, the pristine perovskite film has an onset value of 11.66 eV, the value of the camphor-based film is 12.07 eV, then the Fermi level of pristine film is calculated to be 4.56 eV, compared to that of 4.15 eV for camphor-based film, these two values are consistent with the KPFM measurement shown in **Fig. 3e**.

Fig. R11. UPS spectra of (a) pristine perovskite film and (b) camphor-based perovskite film (under 5 V bias). (This data has been incorporated into **Supplementary Fig. 15**).

12. The authors performed a series of stability tests for perovskite solar cells in different conditions, demonstrating higher stability in bio-molecules modified devices compared to those in pristine ones. As is well known, the stability is determined by not only the perovskite materials but also the charge transport layers. The Li-doped Spiro-OMeTAD generally plays a key factor in the light-heat and/or moisture instability of perovskite devices due to the lower glass-transition-temperature as well as the fast diffusion of lithium ions and the hydrophilicity. Why this work showed such high stability even in harsh conditions (e.g., 1.2 sun, 85% RH, and 85°C)?

Responses: We thank Reviewer #1 for this comment. We agree with Reviewer #1 that the Li-dopant in Spiro-OMeTAD can adsorb moisture and transfer these water molecules into the underlying perovskite and subsequently trigger the perovskite degradation. Here in order to explain the good stability observed in our work, we would like to highlight two aspects: (1) the importance of the hydrophobic reverse micelle layer interfacing the Spiro-OMeTAD and the bottom pristine perovskite and (2) the full Au electrode coverage in our module device.

- (1) As can be seen in **Figs. R12a** and **R12b** below, in the control device, hydrolysis of lithium salts can bring moisture from air to the device, and the degradation of the device is ascribed to the introduced moisture's attack to the bottom perovskite. The Spiro-OMeTAD serves as a moisture collector and transport water molecules into the perovskite thus accelerating the degradation (**Fig. R12a**). In contrast, when a hydrophobic water-proof layer is inserted between the Spiro-OMeTAD and the bottom perovskite, water molecule's attack from Spiro-OMeTAD to perovskite can be significantly alleviated.
- (2) In our stability testing, we employ the module device with design shown in **Fig. R12c**, where there is an Au top electrode holistically covering the whole device. Although tiny open areas can be found in the P3 channel regions, the vast majority of the Spiro-OMeTAD is protected by the Au layer. This can be the reason for the good stability of our devices even under harsh conditions.

Furthermore, we extend our gratitude to Reviewer #1 for astutely spotlighting the potential thermodynamic effects on Spiro-OMeTAD stability. It warrants elucidation that Spiro-OMeTADs are inherently small molecular entities, distinctly disparate from polymers or other covalent network constructs. Consequently, a prototypical Spiro-OMeTAD film, simply doped ionic dopants and devoid of other matrix, it is thought not having the glass-transition. Empirical DSC evaluations on Spiro-OMeTAD proved this assertion, manifesting negligible thermodynamic perturbations below 100°C. However, it's rational to postulate that thermal effect could modulate the hydrolytic equilibrium of the embedded lithium salt. Yet, considering the whole Au coverage in our module with minimized moisture involvement, such assistant thermal effect (85°C) is unlikely to significantly impact the Spiro-OMeTAD.

Fig. R12. Schematic illustrations of how hydrolysis of Li-salt in Spiro-OMeTAD affect (a) pristine perovskite sample and (b) reverse-micelle layer incorporated perovskite sample. (c) A minimodule device we used for our stability testing.

13. The PCE is improved mainly by the increased V_{oc} , why is open circuit voltage improved so much? The authors attributed the reason to the less recombination losses induced by matched energy band arrangement. Any other reasons? More detailed characterization and analysis should be provided.

Responses: We appreciate this comment. The V_{oc} increment can be understood by two scenarios: (1) higher potential energy of excited electrons and (2) reduced recombination (*Energy Environ. Sci.* 2022, 15, 3171-3222). In the main text, we have demonstrated the (2) reduced recombination. Here we would like to elaborate the (1) higher potential energy of excited electrons.

As seen in **Fig. R13**, the higher potential of the electrons in the CB can lead to a higher V_{oc} when these electrons are collected at the electrode. Or in another word, the reverse micelle NP layer has larger bandgap. To verify this, we measure the PL spectra of the bilayer perovskite sample (**Fig. R13b**), exhibiting a blue shift in PL peak compared to the pristine perovskite. Moreover, we fit the PL spectra of the bio-perovskite sample and found that there are two sub-peaks with peak position located at 765 and 784 nm, respectively. The 784 nm is consistent to the PL peak in the pristine perovskite (1.58 eV), which comes from the bottom regular perovskite crystalline grains. And the 765 nm can then be assigned to the top “reverse micelle” biomolecule -perovskite nanocrystal, i.e., a bandgap of 1.63 eV.

Fig R13. PL measurement and band gap estimation. (a) Normalized PL spectra of films of pristine-perovskite and bio-perovskites. **(b)** Two-peak fitting using Gaussian methods, where two sub-spectra peaked at 1.63 and 1.58 eV are obtained, representing the top “reverse micelle” bio-perovskite nanocrystals layer and the bottom regular grain layer, respectively. (This part has been incorporated in **Fig. S18**)

Secondly, we also carried out the C-V measurement on our devices to see if the top bio-perovskite layer can have observable modulating effects on the built-in potential and the overall depletion width. The depletion width can be calculated from equation of (*Appl. Phys. Lett.* 2010, 97, 242501)

$$W = (2\varepsilon_0\varepsilon_r V_{bi}/qN)^{1/2}$$

Where ε_0 is the vacuum permittivity, ε_r is the relative dielectric constant of the perovskites, V_{bi} is the built-in potentials of the device, q is the elementary charge and N is background carrier concentration, respectively. In order to determine the V_{bi} and N , we carried out the Mott-Schottky analysis using the capacitance-voltage (C-V) measurement. **Fig. R14** shows the C^{-2} -V plot, which can be derived to have a correlation of (*Nano Lett.* 2011, 11, 2955-2961)

$$\frac{1}{C^2} = \frac{2(V_{bi}-V)}{A^2 q \varepsilon_0 \varepsilon_r N}$$

Where A is the active area, C is the capacitance, and V is the applied bias, respectively. From the slope in the linear region and the intersect of y-axis, both V_{bi} and N can be determined. The calculated results are presented in **Fig. R14**, where the bio-perovskite device exhibits a built-in potential of 1.18 V with a background carrier concentration of $2.9 \times 10^{15} \text{ cm}^{-3}$, whereas the pristine perovskite device displays a smaller potential of 1.07 V with a concentration of $3.8 \times 10^{15} \text{ cm}^{-3}$. The enlarged V_{bi} is consistent to the observations in the higher V_{oc} in the bio-incorporated device. While the reduced background carrier concentration can be ascribed to a de-doping effect that we assume the biomolecules’ incorporation reduced the self-doping related trap densities, which is consistent to the observation of reduced SRH recombination in our recombination studies. After getting both V_{bi} and N , we further calculate the depletion width at 0 V, which are also presented in the **Fig. R14**. The bio-perovskite device shows a larger depletion width of 938 nm compared to that of 780 nm in the pristine perovskite device. It should be noted that since we use the whole device for the C-V measurement, there are two heterojunctions at the interface of

ETL/bottom perovskite layer and top perovskite layer/HTL (the bottom perovskite layer/top perovskite layer interface may contribute but not as significant as the interface with ETL and HTL, considering the larger energy level off-set at the interface with the charge transfer layers). Therefore, the larger depletion width from the bio-perovskite device is mostly related to a better top perovskite layer/HTL interface, as both devices have identical ETL/bottom perovskite layer interface.

Fig. R14 Mott-Schottky plots (C^{-2} - V) for devices using bio-perovskite and pristine perovskite, respectively. Built-in potential values, background carrier concentrations, and depletion width at 0 V are listed along with the curves. (This part has been incorporated in **Supplementary Fig. 18**)

The relevant part has been added into the **Supplementary Information**.

14. The authors claimed they obtained the highest PCE among MAPbI₃-based devices reported so far. As this value has not been certified, I recommend they tune down the statement.

Responses: We appreciate this comment. The statement is changed to "... one of the highest values for the MAPbI₃-based perovskite solar cells ...".

15. These 12, 20, 23, 24, 27, 31 references' journal abbreviations need correction.

Responses: We appreciate this comment. The corresponding journal abbreviations are corrected.

(12)12. Hou, Y. et al. Homogenization of Optical Field in Nanocrystal-Embedded Perovskite Composites. *ACS Energy Lett.* **7**, 1657-1671 (2022).

(20)28. Hou, Y. et al. Enhanced performance and stability in DNA-perovskite heterostructure-based solar cells. *ACS Energy Lett.* **4**, 2646-2655 (2019).

(23)31. Saraf, R. & Maheshwari, V. PbI₂ initiated cross-linking and integration of a polymer matrix with perovskite films: 1000 h operational devices under ambient humidity and atmosphere and with direct solar illumination. *ACS Appl. Energy Mater.* **2**, 2214-2222 (2019).

(24)17. Jiang, Q. et al. Surface passivation of perovskite film for efficient solar cells. *Nat. Photonics* **13**, 460-466 (2019).

(27)34. Meggiolaro, D., Mosconi, E. & De Angelis, F. Modeling the interaction of molecular iodine with MAPbI₃: a probe of lead-halide perovskites defect chemistry. *ACS Energy Lett.* **3**, 447-451 (2018).

(31)38. Rai, M., Wong, L. H. & Etgar, L. Effect of perovskite thickness on electroluminescence and solar cell conversion efficiency. *J. Phys. Chem. Lett.* **11**, 8189-8194 (2020).

Reviewer #2 (Remarks to the Author):

In this article, the authors select molecules inspired from biological macromolecules and refine their selection through the prediction of solar cell performance parameters. This is a large computational study conducted for a range of biomolecules and their derivatives. The screening process proposed by the authors is interesting for narrowing down the search of additives in perovskite-based solar cells.

The authors propose a reverse micelle formation with the biomolecules and the perovskite crystals. They then apply the materials to the fabrication of photovoltaic devices and characterize the devices.

This article is strong in terms of computational studies, and the photovoltaic device characterization is thorough, but I find that it lacks some experimental evidence to support the proposed molecular additive-perovskite interactions. Nonetheless, the screening performed and the improvement shown through the three tiers of refinement demonstrate the validity of the approach. The degradation study comparing the Hep-modified perovskite and the control is also convincing to demonstrate stabilization effect of the Hep-molecule.

Responses: We express our sincere gratitude to Reviewer #2 for the discerning evaluation of our manuscript. We are also encouraged by your valuable comment that “The screening process proposed by the authors is interesting for narrowing down the search of additives in perovskite-based solar cells”. We concur with the reviewer’s constructive comments on the experimental evidence to support molecular additive-perovskite interaction. To this end, we carry out supplementary investigations in this revision round to bridge this gap and reinforce the manuscript’s conclusions. Once again, we express our gratitude for the constructive feedback and remain committed to addressing the highlighted aspects to ensure the manuscript resonates with both comprehensiveness and rigor.

Please see a few concerns and questions below:

- In some places in the article, I find that the word “biomolecules” has not been used appropriately. At the beginning of the introduction, the authors state “Biomolecules are organic compounds with several types of functional groups such as hydroxyl, methyl, carbonyl, carboxyl, amino, and phosphate groups,¹ playing important role in the formation of molecules like DNA, proteins, carbohydrates, and lipids.”, but this is misleading as this definition of biomolecules appears not to include molecules like DNA, proteins, carbohydrates and lipids, but only the smaller, more primitive molecules. Authors should clearly define which type of small molecules they have used in their studies. In addition, later on, authors mention the “functionalization of hydrocarbons” that can lead to changes in reactivity and interactions – thus, they refer to modified synthetic small molecules here, and not biomolecules. These molecules may be inspired by the structure of biomolecules, but they are not biological. The “Tier 1” molecules that are reported are biological, but the “Tier 3” molecules are not. I request that authors revise their terminology and clarify which types of molecules they used in their studies.

Responses: Thank you very much for this comment and we apologize for this misleading expression. We have revised the beginning sentence into “Biomolecules are organic compounds with several types of functional groups such as hydroxyl, methyl, carbonyl, carboxyl, amino, and phosphate groups,¹ playing important role in the formation of larger organic structures like proteins, carbohydrates, and lipids.”

In our study, we utilized both large and small biomolecules of niacin, β -estradiol, nicotinamide (NAM), DNA, and artemisinin (ART) as the Tier 1 general biomolecules. And in Tier 2, we utilized small biomolecules containing either aldehyde or carbonyl group (i.e., citral, cinnamaldehyde, camphor, menthon, and muscone). While in Tier 3 both biomolecules and reference molecules (including cyclohexanal, heptanal, 7-tridecanone, laurone, hexane and glutaral) were used and tested. In sum, all the Tier 1 and Tier 2 molecules are biological, while tier 3 molecules include both biomolecules and synthetic molecules. We have carefully revised and clarified the corresponding terminology in the manuscript.

Several clarifications have been added to the main text, for example the following statements have been added:

“...the carbonyl grouped alkyl biomolecules as well as its synthetic derivatives...”

“...It should be noted that beyond biomolecules, we also include synthetic molecules in Tier 3 for comparative study and proof-of-concept purposes....”

- Authors refer to the synthesis of perovskites with “molecular additives”, but they have not provided references for previous examples reporting the incorporation of biomolecules with perovskites. They do mention that “certain biomolecules have been reported in prior works”, but no references are provided. This is an entire body of literature that is lacking from the manuscript. For instance: *Small*, 13, 22, 1604305, 2017; *Adv Func Mater* 30, 42, 2005136, 2020 ; *ACS Appl. Nano Mater.* 2022, 5, 9, 12666–12678; and more.

Responses: We greatly appreciate Reviewer #2 guiding us to these valuable literature. In fact, we had incorporated correlated references in our **Supplementary Note 4c**. We understand the importance of these literature and have added them into the main text now.

48. Shih, Y. C. et al. Amino-acid-induced preferential orientation of perovskite crystals for enhancing interfacial charge transfer and photovoltaic performance. *Small* 13, 1604305 (2017).
49. Lang, A. et al. Bioinspired Molecular Bridging in a Hybrid Perovskite Leads to Enhanced Stability and Tunable Properties. *Adv. Funct. Mater.* 30, 2005136 (2020).
50. Aminzare, M., Hamzehpoor, E., Mahshid, S. & Dorval Courchesne, N. M. M. Protein-Mediated Aqueous Synthesis of Stable Methylammonium Lead Bromide Perovskite Nanocrystals: Implications for Biological and Environmental Applications. *ACS Appl. Nano Mater.* 5, 12666-12678 (2022).

- While the others have performed theoretical density of state calculations to determine interactions between the molecules tested and the perovskite lattice, I would have liked to see some experimental evidence of these interactions, for some of the molecules. Several material characterization techniques (spectroscopic techniques in particular) could have been used to validate interactions and to characterize the effects of the molecules of the optoelectronic properties of the perovskites.

Responses: Thank you for this insightful comment. To reveal the interactions between the additive molecules and perovskite materials, we have conducted the XPS measurement on four samples of pristine MAPbI₃ film, Hep-perovskite film, Hex-perovskite film and Glu-perovskite film, as shown in **Fig. R15** (below). **Fig. R15a** shows the XPS spectra of C1s4 (the signal is mainly from the MA cation in perovskite) of different films, where the detailed parameters are listed in **Table R3** (below). The Ref-sample (pristine perovskite) shows a weak CH_x contribution. Samples of Hex and Glu exhibit slightly elevated CH_x contribution, which are consistent to our observation of moderate device performance and non-formation of reverse micelle in the **Tier 3** investigation in the main text. In contrast, the Hep can render a reverse micelle configuration and shows improved device performance. This is also consistent with the significantly elevated CH_x XPS intensity in **Fig. R15a** (the bonding formed between Pb and Hep may affect the MA in the perovskite lattice cage thus leading to the alteration of the chemical environment of MA).

On the other hand, we also measure the XPS spectra of Pb4f in **Fig. R15b** (with the detailed parameters listed in **Table R4** (below)). It should be noted that in the reference sample, there is a strong peak of uncoordinated Pb⁰, which has been observed in prior studies as well (*Adv. Energy Mater.* 2019, 9, 1803766; *ACS Appl. Energy Mater.* 2019, 2, 9, 6624–6633). Intriguingly, by adding different additives, there is a signal decrease of the Pb⁰. Particularly for the Hep sample, the Pb⁰ peak almost diminishes, suggesting the Pb has been coordinated by donating electrons. Moreover, for the Hep-perovskite film, the peak position of the Pb²⁺ has a slightly shift to higher binding regions of 138.67 eV from 138.58 eV of the reference. This can be also ascribed to the formation of molecular interactions between the Pb in perovskite and the functional groups in these biomolecules. These results reveal that there is an interaction between the additive molecules and perovskite.

Fig. R15. (a) XPS C1s4 spectra, (b) XPS Pb4f5 spectra of perovskite thin film without or with hexane, glutaral, heptanal as the additive molecules.

Table R3 Summary of XPS C1s4 spectra of perovskite thin film without or with hexane, glutaral, heptanal as the additive molecules.

Sample	Name	Pos.	FWHM	%Area
Ref	C-N	286.50	1.37	70.21
	CH _x	285.09	1.04	29.79
Hex	C-N	286.50	1.44	68.82
	CH _x	285.07	1.09	31.18
Glu	C-N	286.50	1.42	74.33
	CH _x	285.20	1.03	25.67
Hep	C-N	286.50	1.38	63.83
	CH _x	285.26	1.04	36.17

Table R4 Summary of XPS Pb4f5 spectra of perovskite thin film without or with hexane, glutaral, heptanal as the additive molecules.

Sample	Name	Pos.	FWHM	%Area
Ref	Pb ²⁺	138.58	0.89	67.82
	Pb	136.92	0.82	32.18
Hex	Pb ²⁺	138.58	0.89	75.85
	Pb	136.92	0.85	24.15
Glu	Pb ²⁺	138.62	0.88	82.34
	Pb	136.94	0.84	17.66
Hep	Pb ²⁺	138.67	0.87	92.80
	Pb	137.01	0.97	7.20

- From the main manuscript, it is unclear how the authors have arrived at the conclusion that a reverse micelle was forming. Is there experimental evidence of the formation of these structures?

Responses: Thank you for this question. Please allow us to reintroduce the underlying mechanism for this reverse micelle formation and the resultant bilayer structure.

Colloidal-crystallization system deviating from traditional colloidal fundamentals: In this work, the precursor solution is a complex system. Specifically, we have ionic precursors of perovskite (e.g., MA⁺, Pb²⁺, I⁻), intermediate states after assembly of certain precursors (e.g., MA-Pb-I intermediate clusters), isolated biomolecules, and biomolecule-adsorbed clusters (possible components in the colloidal system is shown in the following **Fig. R16**). This is different from classic colloidal system consisting of a liquid continuum and solid, submicrometric particles which are stable within the particle itself. This difference makes the complex co-existence of a colloidal process and an additional crystallization process in our system. Competitive and/or synergistic and/or coupling effects between these two main processes can be presented in the system, which leads to five possible pathway branches as shown below.

Fig. R16. Possible components in the co-colloidal-crystallization system containing perovskite precursors and biomolecules. Underlying fundamentals in this complex system is regarding the interaction between crystallization and colloidal dynamics, in-depth scientific insights that can be of interest in future research.

Unexpected ‘perovskite/perovskite bilayer’ and its benign electrical feature:

Continuing the discussion on kinetics, one unexpected discovery in this work is that certain biomolecules can lead to this bilayer structure. We present one possible mechanism as below:

The co-existence of multiple components in **Fig. R16** can lead to many hypothetical manifestations and eventually various film morphologies and topographies. In this work, we tested a lot of different conditions and processing parameters, and eventually discovered that certain biomolecules can lead an interesting bilayer perovskite structure. In competition to other pathways shown in **Fig. R16**, **Fig. R17** shows one branch to give rise to a bilayer structure, where we found the top biomolecule-wrapped perovskite nanocrystal – reverse micelle structure – displaying higher energetic states beneficial to high V_{oc} in the device.

Fig. R17 Hypothetical formation mechanism of the perovskite/perovskite bilayer structure where the top perovskite nanocrystals exhibit higher energy states whereby leading to a higher potential to the device.

To further investigate the reverse micelle formation during the perovskite precursor solution, we have conducted dynamic light scattering (DLS) measurement. As shown in **Fig. R18**, the pristine perovskite solution contains the colloid with a size of around 1.12 nm (distributed from 0.62 nm to 2.01 nm), which represents the small metastable aggregation not yet forming the crystal in the solution. In comparison, the camphor-perovskite not only presents this peak around 0.83 nm (distributed from 0.62 nm to 1.12 nm) but also a secondary peak around 142 nm (distributed from 122 nm to 190 nm), demonstrating the formation of the larger reverse micelle in the solution system.

Fig. R18. DLS spectra of pristine perovskite solution and 3 mg/mL camphor-perovskite solution.

Additionally, in the revised **Supplementary Information (Supplementary Fig. 14, cited below)**, we also incorporated the following information and correlated discussions. Particularly, **Supplementary Figure 14a** shows the Top-view SEM image of MAPbI₃ film doped with 3 mg/mL camphor and **Supplementary Figure 14e** shows the elemental mapping of the MAPbI₃-camphor nanocrystal: where I and Pb are exclusively from perovskite while C is from both perovskite and camphor. The yellow dash line denotes the perovskite grain edge, while the blue dash line denotes the periphery of the particle. It can be seen that the C has a larger region than the Pb, suggesting the carbon-dominant organic (camphor) is wrapping up the perovskite core.

In addition to this, the significantly improved hydrophobicity (**Fig. 3b**) also validates the biomolecular wrapping effect. And consequently from all of above discussions, we converge to the concept of reverse micelle nanoparticle.

Figure S14 (a) Top-view SEM image of MAPbI₃ film doped with 3 mg/mL camphor. (b) Schematic of nanocrystalline reverse micelle structure. (c) Illustration of perovskite-camphor reverse micelle structure (a half micelle is shown here to visualize the inner perovskite). (d) High-angle annular dark field (HAADF) image of perovskite-camphor nanoparticle. (e) Elemental mapping of the MAPbI₃-camphor nanocrystal: where I and Pb are exclusively from perovskite while C is from both perovskite and camphor. The yellow dashed line denotes the perovskite grain edge, while the blue dashed line denotes the periphery of the particle. Electron energy loss spectroscopy (EELS) comparing perovskite-biomolecule vs. pristine-perovskite, to

validate the chemical interaction from biomolecules to the Pb element in the perovskite. Briefly, the Pb M2 and M3 edges display a large shift after biomolecular-adsorption in the sample, which is in contrast to the case of pristine MAPbI₃. This means that there is a significant change in the coordination environment around Pb atom, which can be ascribed to the Pb-carbonyl group bonding between perovskite and the biomolecule. (f) Possible components in the co-colloidal-crystallization system containing perovskite precursors and biomolecules. Underlying fundamentals in this complex system are regarding to the interaction between crystallization and colloidal dynamics, in-depth scientific insights that can be of interests in future research. (g) Hypothetical formation mechanism of the perovskite/perovskite bilayer structure where the top perovskite nanocrystals exhibit higher energy states whereby leading to a higher potential to the device.

- The authors state that after 5mg/mL, the additives had negative effects on device efficiency and postulate that it is due to their insulating nature. Have they investigated what happens more specifically? Above a certain concentration, do the molecules still interact in the same manner with the perovskite crystals? Do they form structures separate for the perovskite? Do they accumulate in sections of the films?

Responses: We thank you for this comment. Here we provide the following mechanisms schematized in Fig. R19 coupled with cross-sectional SEM images (Fig. R20) supporting it. Specifically in condition of a small number of biomolecules (Fig. R19a), it is not enough to form a complete “reverse micelle” NP layer to cover the bottom large grain film. The transport is in good condition, but moisture can also enter the bottom perovskite. While in an ideal case (Fig. R19b) with suitable concentration of biomolecules, a thin and complete layer of “reverse micelle” NP forms which will not only secure the blocking of moisture’s attack but also offer a good transport. It should be noted that although the biomolecular ligand at the surface of “reverse micelle” NP is less conductive, there could be the possibility of charge tunnelling/leaking between the inner perovskite core and the outer transport layer or the neighboring perovskite crystals. However, while at higher concentrations (Fig. R19c), not only the tunnelling probability drops due to the thicker “reverse micelle” NP layer and more insulative ligand barriers but also the presence of voids and film discontinuity can increase the series resistance for transport. This morphological change is well evidenced in Fig. R20. The thickness of the top layer is increasing with the enlarged concentration, from 80 nm at 0.5 mg/mL, 100 nm at 1.0 mg/mL, 125 nm at 3.0 mg/mL, 195 nm at 5.0 mg/mL, 250 nm at 10.0 mg/mL, to 405 nm at 20.0 mg/mL. It should be noted that at higher concentrations of 20 mg/mL, not only the top layer presents voids but also the bottom layer shows significantly reduced crystal grains and higher density of voids. All of these can lead to a performance drop in solar cells.

Fig. R19 Scheme showing how the concentration of biomolecule can affect the morphology and its hypothetical effects on both charge transport and moisture stability. (a) Addition of too less biomolecules cannot form a complete “reverse micelle” NP layer and the exposed top region of the pristine large grain can be attacked by moisture, while the transport is good. (b) An ideal condition where both transport and moisture resistance are in good condition. (c) Addition of too much amount of biomolecules can lead to morphological change significantly, leading to the formation of disordered and smaller crystals in the bottom.

Fig. R20. Cross-section SEM images of perovskite thin films with different concentrations of heptanal molecules as a support for the proposed morphological manifestations upon different concentrations of biomolecules (Fig. R19).

- The text should be reviewed carefully for typos and a few grammatical mistakes. For instance, from the first paragraph only of the results section: “the halide perovskite is A metastable ...” “the merngence of point defectS, ...” “material degradation particularLY” “In natural systemS,...” ...

Responses: We are grateful to Reviewer #2 for the meticulous review and for highlighting the grammatical errors. We have thoroughly reviewed and amended the manuscript to rectify the typos and grammatical inconsistencies.

“In principle, the halide perovskite is a metastable crystal with weakly bonded interaction within their octahedral unit, and the easy atomic lattice-site activation can lead to the emergence of point defects, multilevel traps, and even the lattice collapse to trigger the material degradation particularly in the presence of moisture.”

“In natural systems, there are many cellular processes occurring in aqueous environment, which are modulated by the hydrophobic and hydrophilic functional groups within biomolecules.”

Reviewer #3 (Remarks to the Author):

In this manuscript, three tiers of biomolecules have been systematically tested and the results reveal that “reverse-micelles” consisting of biomolecules and precursors are involved in the perovskite crystallization. As a result, the core@shell perovskite@biomolecule nanoparticles are accumulated on the top of the perovskite film, leading to Fermi level splitting within the perovskite film and suppressed reverse charge transfer. By this reverse micelle strategy, excellent device performance could be achieved for either small-area or large-area solar cells or modules. This work deserves publication in nature communications. Nevertheless, several issues need to be clarified before acceptance.

Responses: We thank Reviewer#3 for your time and patience reviewing this work. We extend our gratitude for your thorough review and insightful comments. Your constructive feedback has been instrumental in enhancing the quality of this manuscript. In response to your suggestions/questions, we have undertaken additional experiments and discussions. And we believe that these revisions have substantially enriched the manuscript.

1. The introduction focuses on the interaction between metallic compounds and bio-molecules in nature, which is too general to get the real point. The introduction shall focus on the recent advances of various perovskite solar cells with bio-molecules and point out the problems or challenges in this field.

Responses: We thank Reviewer #3 for this comment. In response to the comment, we have streamlined our introduction to highlight the advancements in perovskite solar cells that integrate biomolecules. Indeed, recent research in perovskite photovoltaics has unveiled the multi-faceted roles of biomolecular/organic dopants, with their influences spanning across aspects such as device efficiency, lifetime, hysteresis, self-cleaning, self-healing, etc. The following **Fig. R21** (top panel) lists examples of prior works, which mostly have a specific focus on either defect passivation, or crystallization modification, or others, but *oversighting other possible effect in the meantime, for example, how the additive can affect at different length scales such as meso- or interfacial level*. While in the real case (**Fig. R21** bottom panel), many effects occur at different length scales and can be either synergistically or contradictorily influence the overall device performance. Consequently, to deliver a holistic understanding of the role of bio-additives in perovskite solar cells, it becomes imperative to recognize their multifaceted influences, not just at the molecular level but also at the grain, interface, and device levels. This hierarchic influence becomes pivotal when gauging the overall device performance under real-world conditions.

Furthermore, the device performance manifested in different scenarios (either lifetime, or efficiency, or other figure-of-merits such as hysteresis, functionalities of self-healing etc.) can also be affected significantly different between each other. In other words, the ‘hidden Blackbox’ linking the method (e.g., introduction of a bio-additive) to the results (i.e., device performance) can be an interpenetrating network consisting of various mechanisms at different length scales. Therefore, simply focusing on a singular perspective (e.g., passivation at molecular level) without considering grain-, interface-, device-level effect, can be less convincing to explain the real device performance in practical cases. There will need a comprehensive understanding of how the additive can trigger what exact effect at different length-scales, and how these different effects can lead to different changes in all figure-of-merits of the device performance matrix.

We think this comprehensive understanding of a bio additive’s effect at different length scales is of great scientific importance to visualize the ‘hidden box’. And in this work, we study this hierarchic series effects from molecular scale (DFT, HRTEM, XRD), to grain level (AFM, KPFM), to interface level (SEM), to small device level with scenarios of the photoelectrical (electroluminescence), optical (PL), electrical (SRH loss study and depletion region analysis), to module level and practical level evaluation (various stability testing). We believe this multi-scale approach offers a more robust understanding of the bio-additive’s integration into perovskite solar cells, thereby addressing the central challenges in this domain.

In fact, different mechanisms can trigger a series effect throughout length-scales from molecular level to meso- and interface- levels

- There lacks a comprehensive vision: how a bio-additive can trigger the what chain effect over different length scales?
- There lacks an in-depth understanding: how the nominal performance enhancement is correlated to what fundamental science?
- Discovery of new colloidal manifestation in crystalline system – ‘reverse micelle’ crystal
- Down-selection material screening leads to a ‘general rule of selection’

Fig. R21 Background information. Top panel: examples of prior works using additives and mostly introduce one specific mechanism to explain the device improvement. However, in real case, many effects at different length scale (bottom panel) can interactionally lead to different influence on device. Furthermore, their individual, synergistic, and contradictory effects on different device features of stability, efficiency, and other factors can also vary from one to one. Overall, device performance can be very complexly affected from an additive. Elucidating this ‘hidden Blackbox’ linking the additive and the device results will not only need comprehensive understanding of one molecular effect over the whole length scale but also the verification of different molecules of the same type. That is, matrix-type research, which is exemplified by this work.

For clarity, we have incorporated a succinct discussion in the introduction that underscores the recent research on perovskite solar cells integrated with biomolecules. This addition also sheds light on the pertinent challenges and issues in this direction. The included details are as follows:

“Fundamentally, the biomolecules can be classified into finite groups in terms of their functional chemical structures, which are intrinsically responsible for multiple levels of biochemical activities. Prior investigations into the integration of biomolecules with perovskite have delineated a range of mechanisms, from dangling bond¹⁴/lattice terminal¹⁵/point-defect¹⁶/surface passivation¹⁷, to protective barrier introduction¹⁸, crystallization modification¹⁹, and grain boundary²⁰/interface engineering²¹, etc. However, most of these studies tend to emphasize a singular mechanism, potentially neglecting concurrent influences. Practically, mechanisms operating at varied scales

can either synergistically or contradictorily interact with each other, leading to versatile holistic manifestations at device level. For instance, parameters like device lifetime, efficiency, and other performance metrics could be differentially impacted by these mechanisms. This underscores the intricate interplay between the incorporation of an additive and the consequent device performance, which can be mediated through a myriad of mechanisms across diverse scales. Hence, a comprehensive understanding, stretching from molecular intricacies to broader grain, interface, and device dimensions, is paramount to decipher the hierarchical impacts of biomolecules on device performance aspects. Adopting such a multi-scale perspective could provide a more in-depth comprehension of the incorporation of biomolecules into perovskite solar cells, directly addressing the central challenges in this field. Considering the extensive variety of biomolecules and the multi-dimensional nature of their impacts, strategic research design coupled with judicious molecular selection becomes essential.”

2. The authors mentioned the motivation of this work in the results section. That is, “there lacks a systematic understanding on how a molecule can induce various effects at different length scales, as well as how different”. I suggest providing the motivation in the introduction section for better readability.

Responses: We thank this comment. Following the last response, the detailed description of motivation has been well incorporated in the introduction.

Here, please allow us to mention once again that simultaneously studying all the effects and elucidating the inter-influenceable network is important to understand the holographic mechanism matrix. Each independent work in prior research gives us a localized vision that certain types of molecules can positively affect the device performance. But why those molecules are workable and how the seemingly different molecules can have similar effects on device performance? To answer these questions, it will need a systematic and consistent set of investigations to uncover any hidden rules of material selection. Meanwhile many contradictory statements from different reports can then be explained based on these hidden rules, which we think can help the field move forward. In this work, we would like to solve these problems following two directions: (a) from ‘vertical direction (length-scale)’, we carry out studies from molecular level to meso-level, ..., all the way to module level; (b) from ‘horizontal direction (expanding from one prototype biomolecule and systematically investigate the similar molecules’ effect), we select representative molecules with rational tier hierarchic classification to comprehensively find a rule of material selection.

Overall, we appreciate Reviewer #3 for this constructive suggestion, and we have well integrated this new part in the introduction.

3. The new point in this manuscript is to establish the mechanism on how the “reverse-micelle” affects perovskite crystallization and its performance. The authors shall provide a more clear definition of the term “reverse micelle” in this manuscript and identify the size distribution of reverse micelles based on different bio-molecules.

Responses: Thank you very much for this insightful comment. We have incorporated updated information as well as the clear definition of “reverse micelle” additive-perovskite nanocrystal in the revised file. Particularly in **Supplementary Note 5**, we have updated the definition, formation mechanism in comparison to classic colloidal system, and extended discussion on the molecular origin.

This information include:

“Supplementary Note 5 Formation mechanism of “reverse micelle” and “bilayer”

Reverse micelle: A “Micelle” is an aggregate (or supramolecular assembly) consisting of surfactant molecules (e.g., phospholipid molecules typically in bio-cell membrane). They form associated colloidal system in a typical case of surfactants and water system.⁵² The functional molecules share the similarity of a hydrophilic “head” and hydrophobic “tail”, as shown below.

And in aqueous condition, the hydrophilic “head” tends to contact with the polar water solvent, while the hydrophobic “tail” attracts each other. This can lead to certain microstructures, including micelle models (head facing outside and tails are oriented toward the center of the sphere), lamellar models, and rod models. These various manifestations are driven by the combination of different molecular interactions present in the system, mainly including the (i) tail-tail interaction and (ii) head-solvent attraction.⁵³

In our case the solution systems incorporate both these amphiphilic molecules and the perovskite precursors, leading to complex ingredients. Particularly the presence of the interaction between the perovskite and the additive molecules (Pb-“head” interaction) triggers the formation of a “reverse micelle” structure. We tested the dynamic light scattering (DLS) measurement on different samples and observed a secondary peak at a larger size region (appeared at a size level of 100s nm), in distinct contrast to the pristine samples that only exhibit a mono-peak at 1s nm level.

This formation driving force can be understood by the molecular interaction between the carbonyl head and the perovskite lattice thoroughly present during the wet film drying process. This molecular attraction will lead to the biomolecules adsorbed onto the perovskite surface, with the hydrophobic tail of the biomolecules facing outside, which is also the reason for the higher hydrophobicity of the film (as verified by contact angle and water dropping tests in this study). However, formation of such a “reverse micelle” is not easily accessible and requires several conditions. We have shown there are five possible pathways leading to different mechanisms in **Supplementary Fig. 14f**. Not all the additive biomolecules can give rise to such a “reverse micelle” structure. In this study, we extensively investigate this unexpected phenomenon through a down-selective screening methodology.

Bi-layer: ...”

For the question on the size distribution of different molecules, we have added the dynamic light scattering (DLS) measurement to identify the colloidal size in the solution. Taking camphor and heptanal as an example, we measured the DLS spectra of solution samples of pristine-perovskite, camphor-perovskite, and heptanal-perovskite of different additive concentrations of 1 mg/mL, 3 mg/mL, and 5 mg/mL. **Fig. R23** shows the DLS spectra. The results reveal that the pristine perovskite solution exhibits a mono-peak around 1.12 nm (distributed from 0.62 nm to 2.01 nm), suggesting that there is only one type of small metastable aggregation not yet forming the crystal in the solution. In contrast, with incorporation of camphor or heptanal additives, a secondary peak at larger length scale (>100 nm) has been observed. Specifically, by increasing the concentration from 1 to 3 and 5 mg/mL, the colloidal size increases from ~106, to ~142 nm and ~295 nm in the camphor-perovskite solution. Similarly, by increasing the concentration from 1 to 3 and 5 mg/mL, the colloidal size increases from ~142, to ~190 and ~459 nm in the heptanal-perovskite solution. These larger size aggregations suggest that the reverse micelle has been formed in the solution, supporting our hypothetical mechanisms proposed in **Fig. 1c**.

Fig. R23. (a) DLS spectra of pristine perovskite solution and camphor-perovskite solution with 1, 3, 5 mg/mL doping concentrations. (b) DLS spectra of pristine perovskite solution and heptanal-perovskite solution with 1, 3, 5 mg/mL doping concentrations.

4. The authors added acetonitrile into the ethanol solution of perovskite precursors. Please explain why acetonitrile and ethanol are used here. In fact, DMSO, DMF, and butyrolactone are more common solvents for perovskite precursors.

Responses: We appreciate the insightful question. Indeed, the DMF/DMSO solvent system has traditionally been the mainstay for perovskite precursor solutions. However, a challenge we faced was the limited solubility of most biomolecules within this system. Drawing upon our previous work (*Energy Environ. Sci.* 2020, 13, 3412-3422; *Joule*, 2020, 4, 615-630), we elected to employ a solvent blend of methylamine-containing ethanol (Eth-MA) and acetonitrile (ACN). In this chosen system, we observed that biomolecules dissolved more favorably in Eth-MA.

In the meantime, another significant advantage offered by these solvents is their low boiling points, enabling swift perovskite film crystallization at ambient conditions, thereby negating the need for thermal annealing, as detailed in (*Matter* 2021, 4, 775-793). This particular characteristic is advantageous for the synthesis of biomolecular-engineered perovskite films where otherwise a much higher annealing temperature can generate risks on de-activating or even degrading the biomolecules.

Lastly, the processing simplicity (exemption of thermal annealing, antisolvent treatment, etc.) minimizes variability in manufacturing. This allows for a more precise evaluation of the introduced biomolecules' effectiveness.

5. In Figure 1a, ball-stick models are provided for bio-molecules, but the problem is poor readability. Chemdraw structure would be better here for higher clarity.

Responses: We thank this comment. We have replaced the ball-stick models with the chemical structures drawn by Chemdraw. Here is the replaced figure partially.

6. In Figure 1a, DNA was classified into the class with PO₄ group. Nevertheless, there are also amino and amide groups in DNA. In NAM, there is not only amino group but also -CO- group. So, why DNA or NAM is classified into the class with PO₄ or amino? Please explain why PO₄ or amino group has superiority when considering its classification.

Responses: We appreciate this insightful comment. DNA and NAM are in the Teir 1 molecules, most of which are selected from prior reports but previously being used in different perovskite compositions or device configurations. According to the literature, both the DNA and NAM have positive effects on device efficiency and meanwhile they have the PO₄ and amino groups, respectively. In this study, we compare them in the same condition using identical perovskite composition and device structure.

In terms of the case that the DNA and NAM having more than one functional group. We firstly consider the reaction priority between functional groups in the same molecule, by calculating the energy difference before and after adsorption of one group to the perovskite by DFT. Taking NAM as an example, the NAM has both amino group and carbonyl group. We utilized the DFT to reveal the energy difference in both cases. **Supplementary Fig. 4** shows the electron density difference before and after the adsorption of the NAM biomolecules onto the surface regarding either amino group or carbonyl group. The amino group's interaction with the perovskite yields an adsorption energy of -0.89 eV, which is substantially more favorable compared to the -0.16 eV associated with the carbonyl group. This underscores the amino group's predominant role in influencing perovskite crystallization in the presence of NAM.

The scenario with DNA is more intricate. DNA comprises four nucleobases: adenine (A), cytosine (C), guanine (G), and thymine (T). The interactions between each nucleobase and the perovskite warrant individual attention. Our DFT results, presented in Fig. S5, elucidate the energy variations for each nucleobase upon interaction with the perovskite. Moreover, given the size and complexity of the DNA

molecule, steric hindrance can significantly influence its orientation relative to the perovskite (*Nature Education* 2008, 1(1), 100). Consequently, we postulate that the outermost phosphate group at the DNA helix edge would be more inclined to interact with the perovskite due to minimized steric constraints.

We hope that these analyses provide clarity on our rationale for designating DNA and NAM as **Tier 1** biomolecules.

7. Fig. 1c shows the film drying process where the “reverse micelle” structures are firstly assembled in the solution state, accumulating at the top surface of the wet membrane. However, there lacks direct evidence to support this proposal. In Figure 3d, the HRTEM image reveals there is an amorphous layer on the surface of the perovskite. To verify this amorphous layer being from camphor, ToF-SIMS and XPS analysis shall be provided. To verify the formation of reverse micelle in the precursor solution, light dynamic scattering analysis shall be provided. Is it possible that the reverse micelles also accumulate at the buried interface of perovskite film (the bottom side of perovskite film)? The authors shall consider or exclude such a possibility by depth-dependent analysis (i.e., ToF-SIMS).

Responses: We thank this comment and your rational concerns on the chemical insights of our “reverse micelle” model, as well as advising us these important characterizations. Accordingly, we have supplemented these characterizations and provided the following discussion related to these scenarios.

XPS validation of molecular interaction: To reveal the interactions between the additive molecules and perovskite materials, we have conducted the XPS measurement on four samples of pristine MAPbI₃ film, Hep-perovskite film, Hex-perovskite film and Glu-perovskite film, as shown in **Fig. R24** (below). **Fig. R24a** shows the XPS spectra of C 1s (the signal is mainly from the MA cation in perovskite) of different films, where the detailed parameters are listed in **Table R5** (below). The Ref-sample (pristine perovskite) shows a weak CH_x contribution. Samples of Hex and Glu exhibit slightly elevated CH_x contribution, which are consistent to our observation of moderate device performance and non-formation of reverse micelle in the **Tier 3** investigation in the main text. In contrast, the Hep can render a reverse micelle configuration and shows improved device performance. This is also consistent with the significantly elevated CH_x XPS intensity in **Fig. R24a** (the bonding formed between Pb and Hep may affect the MA in the perovskite lattice cage thus leading to the alteration of the chemical environment of MA).

On the other hand, we also measure the XPS spectra of Pb 4f in **Fig. R24b** (with the detailed parameters listed in **Table R6** (below)). It should be noted that in the reference sample, there is a strong peak of uncoordinated Pb⁰, which has been observed in prior studies as well (*Adv. Energy Mater.* 2019, 9, 1803766; *ACS Appl. Energy Mater.* 2019, 2, 9, 6624–6633). Intriguingly, by adding different additives, there is a signal decrease of the Pb⁰. Particularly for the Hep sample, the Pb⁰ peak almost diminishes, suggesting the Pb has been coordinated by donating electrons. Moreover, for the Hep-perovskite film, the peak position of the Pb²⁺ has a slightly shift to higher binding regions of 138.67 eV from 138.58 eV of the reference. This can be also ascribed to the formation of molecular interactions between the Pb in perovskite and the functional groups in these biomolecules. Additionally, the XPS spectra of the I 3d of Hep-perovskite also exhibits a highest degree of binding energy shifting (**Fig. R24c**). These results reveal that there is an interaction between the additive molecules and perovskite.

Rever micelle validation by dynamic light scattering (DLS): To further investigate the reverse micelle formation during the perovskite precursor solution, we have conducted DLS measurement. As shown in **Fig. R24d**, the pristine perovskite solution contains the colloid with a size of around 1.12 nm (distributed from 0.62 nm to 2.01 nm), which represents the small metastable aggregation not yet forming the crystal in the solution. In comparison, the camphor-perovskite not only presents this peak around 0.83 nm (distributed from 0.62 nm to 1.12 nm) but also a secondary peak around 142 nm (distributed from 122 nm to 190 nm), demonstrating the formation of the larger reverse micelle in the solution system.

Time-of-Flight Secondary Ion Mass Spectrometry (ToF-SIMS) investigation: The ToF-SIMS measurement is carried out to study the depth profile of different elements in both the pristine perovskite and Hep-perovskite films. As seen from **Fig. R24e** (pristine perovskite) and **Fig. R24f** (Hep-perovskite), there is an obvious difference of the depth profile of carbon. Particularly at the initial 100 nm region, the Hep-perovskite shows a high atomic ratio of carbon, suggesting that most Hep organic molecules are located at the top surface of the sample. In contrast, in the pristine-perovskite the atomic ratio of carbon is

consistently low compared to other elements, which can be understood by the uniform composition throughout the film in thickness direction and the relatively much smaller atomic mass of carbon compared to lead and iodine.

These results demonstrate that the reverse micelle layer is exclusively present on the top-surface of the perovskite film.

Fig. R24. XPS spectra of (a) C1s4, (b) Pb4f5, and (c) I3d5 of perovskite film without or with additive molecules of hexane, glutaral, and heptanal. (d) DLS spectra of pristine perovskite solution and 3 mg/mL camphor-perovskite solution. (e) ToF-SIMS spectra of pristine perovskite film. (f) ToF-SIMS spectra of Hep-perovskite film under the doping concentration of 3 mg/mL.

Table R5 Summary of XPS C1s4 spectra of perovskite thin film without or with hexane, glutaral, heptanal as the additive molecules.

Sample	Name	Pos.	FWHM	%Area
Ref	C-N	286.50	1.37	70.21
	CH _x	285.09	1.04	29.79
Hex	C-N	286.50	1.44	68.82
	CH _x	285.07	1.09	31.18
Glu	C-N	286.50	1.42	74.33
	CH _x	285.20	1.03	25.67
Hep	C-N	286.50	1.38	63.83
	CH _x	285.26	1.04	36.17

Table R6 Summary of XPS Pb4f5 spectra of perovskite thin film without or with hexane, glutaral, heptanal as the additive molecules.

Sample	Name	Pos.	FWHM	%Area
Ref	Pb ²⁺	138.58	0.89	67.82
	Pb	136.92	0.82	32.18
Hex	Pb ²⁺	138.58	0.89	75.85
	Pb	136.92	0.85	24.15
Glu	Pb ²⁺	138.62	0.88	82.34

	Pb	136.94	0.84	17.66
Hep	Pb ²⁺	138.67	0.87	92.80
	Pb	137.01	0.97	7.20

8. As illustrated in Figure 1a, the reverse micelle is formed involving both bio-molecular and perovskite precursors, which is reasonable. So, again I suggest that light dynamic scattering analysis shall be performed to identify the size distribution of reverse micelle in precursor solution.

Responses: We thank again for this comment. As for the size distribution of different molecules, we used camphor and heptanal as the testing samples for the dynamic light scattering (DLS) measurement under different concentrations like 1 mg/mL, 3 mg/mL, and 5 mg/mL, with comparison to the pristine perovskite. **Fig. R23** (in our response to your 3rd comment, shown above in this letter) shows the DLS spectra. The results reveal that the pristine perovskite solution exhibits a mono-peak around 1.12 nm (distributed from 0.62 nm to 2.01 nm), suggesting that there is only one type of small metastable aggregation not yet forming the crystal in the solution. In contrast, with incorporation of camphor or heptanal additives, a secondary peak at larger length scale (>100 nm) has been observed. Specifically, by increasing the concentration from 1 to 3 and 5 mg/mL, the colloidal size increases from ~106, to ~142 nm and ~295 nm in the camphor-perovskite solution. Similarly, by increasing the concentration from 1 to 3 and 5 mg/mL, the colloidal size increases from ~142, to ~190 and ~459 nm in the heptanal-perovskite solution. These larger size aggregations suggest that the reverse micelle has been formed in the solution, supporting our hypothetical mechanisms proposed in **Fig. 1c**.

9. The authors claim that there is Fermi level splitting within the perovskite film, which will enhance the built-in electric field in the regular structured device. To confirm this conclusion, disproof shall be provided. For example, when the camphor doped perovskite film is applied to inverted device, the Fermi level splitting will weaken the built-in electric field and lower the device performance.

Responses: We thank this insightful comment. Here we would like to (1) firstly introduce the mechanisms on how the additive change the perovskite's electronic band structure and **theoretically** illustrate how it elevate V_{oc} in n-i-p configuration, and (2) provide the disproof of lower V_{oc} in p-i-n configuration **experimentally**.

(1) The bio-modulation on electronic band structure:

In ideal case (infinite periodic lattice), the halide perovskite material has a direct band feature with (i) Pb's p orbital contributing the CBM (conduction band minimum), (ii) I's p and Pb's s orbitals contributing the VBM (valance band maximum), (iii) there is no states located within the forbidden band ('cell-2b' in the following matrix in **Fig. R25** shows a DOS of MAPbI₃ based on prior literature,^{9,10} and a DOS illustration is presented in 'cell-3b'). Considering the photo-excited condition, excited electrons can occupy the orbitals in the CB but mostly go to the CBM through an ultrafast thermalization process and eventually lead to a distribution schematized in the 'cell-5b' (**Fig. R25**). **In real case ('row c')**: the terminal Pb can have significant contribution to the DOS. Unit 'cell-2c' shows an example of the terminal Pb's DOS from first-principles calculation, where a high-density deep states located within the forbidden band is presented ('cell-3c'), which can be ascribed to the unbonded p orbital of Pb. This deep states have also been observed and reported in prior literature,¹¹ suggesting its general presence in perovskite materials. Unit 'cell-4c' shows the occupying of these deep states, along with those tail trap states (shallow trap sates) which eventually lead to the distribution of the excited states of electrons centered below the CBM ('cell-5c'). Due to the wide existence of both shallow trap states and lattice terminal, this lower excited state distribution center is believed to be a general case in real perovskite films. **In our case ('row d')**: From various DFT results in multiple conditions in our **supplementary information**, we found that the deep states from terminal Pb can be eliminated by certain biomolecular modification ('cell-2d', schematized in 'cell-3d'). In the meantime, we also observed lower shallow trap states due to the passivation effect. These two effects jointly lead to a higher energy level centered excited electron distribution, as shown in 'cell-5d'.

Overall, from these conclusions the exemption of terminal Pb deep states leads to an up-dragging of electron DOS and eventually lead to higher Fermi level and excited states.

Fig. R25. Matrix comparison showing in different cases, how terminal Pb's DOS can affect distribution of excited electrons in the material, and how the biomolecules can modulate this effect. We use notation of 1,2,3,4,5 to denote the column and a,b,c,d to denote the row, so that each cell in this matrix figure can be referred for discussion in above narrative. And Energy level diagram of perovskite layer and spiro-OMeTAD layer before and after contact, adapted from our **Supplementary Fig. 16** is **Supplementary Information**.

Contribution to more efficient electron blocking effect between perovskite/HTL interface: As seen below, in light condition, Fermi levels of electron and hole are separated due to the dynamic equilibrium condition. Similar to dark condition, holes can still have an ohmic contact and thus freely transfer across the interface. For electrons, there exist the Schottky barrier, with a value of $\phi_{B,e} = \phi_{HTL} - \phi_{p,e}$. Here the $\phi_{p,e}$ is corresponding to the quasi-Fermi level of electron under illumination. Hence, a higher quasi-Fermi level of electron, i.e., a smaller $\phi_{p,e}$ can lead to larger $\phi_{B,e}$, which is beneficial for blocking the electron to transport towards HTL (i.e., blocking effect for reverse electron transfer).

(2) disproof of lower V_{oc} in p-i-n configuration

It should be noted that the above discussion is based on n-i-p structure, where we focus on the perovskite/HTL interface because the reverse micelle layer is on top. As Reviewer #3 pointed out, in a p-i-n device, if such an effect is reversely impacting the device remains unclear. We appreciate this insightful thought and in order to verify as a disproof, we fabricated the corresponding device with a structure of ITO/PTAA/MAPbI₃/C60/BCP/Ag. And we measured the device performance as shown in **Fig. R26**. The control device has a V_{oc} of 1.076 V, a J_{sc} of 22.33 mA/cm², an FF of 78.8 % and a PCE of 18.92 %. However, after the addition of 3 mg/mL heptanal, the device performance drops, with a V_{oc} of 1.048 V, a J_{sc} of 22.67 mA/cm², an FF of 71.6 % and a PCE of 17.00 %. In this way, our reverse micelle method is not suitable for the p-i-n perovskite solar cells. It seems in this result, the Hep-device shows a smaller V_{oc} , acting as a disproof.

Fig. R26. J-V characteristics of p-i-n device without or with 3 mg/mL heptanal additive molecules.

10. In Figure 3a, the camphor modified devices deliver the largest V_{oc} , while its PCE is not the highest. Please discuss how the various reverse micelles affect the FF and J_{sc} to varying degrees.

Responses: We appreciate this comment. The concentration and type of biomolecules can lead to different reverse micelle/pristine perovskite bi-layer configurations as well as the electrical feature of it. This can have different impacts to the figure-of-merits of PV performance. Among them, the thickness of the top layer is an important factor. When examining **Fig. R27a** (shown below), which displays the cross-section SEM images of perovskite films fabricated with all the Tier 2 biomolecules (all at a concentration of 3.0 mg/mL), we can discern the thickness of the top layer corresponding to each molecule. This is further quantified in **Fig. R27b** as well. Among these, the film developed with camphor exhibits the thinnest (lowest magnitude) and most uniform top layer (smallest deviation).

We then compare the thickness of the top layer of reverse micelle nanoparticles with V_{oc} , FF, J_{sc} and PCE of the corresponding device. It is evident that the PCE and J_{sc} have a similar inverse correlation with the top layer thickness. This can be understood by the thickness-dependent electrical resistance effect by the reverse micelle nanoparticle top layer on the solar cell. Empirically, the larger thickness of the top layer, the more insulative (because of the larger density of voids in the nanoparticle packing), the more charge carrier losses during transport, and thus the lower J_{sc} . For the V_{oc} and FF, it is seen that all the additive modified devices exhibit higher V_{oc} and slightly lower FF compared to the control. We have

discussed the higher V_{oc} in the main text and the lower FF can be ascribed to the inferior transport feature due to the insulative organics in the material. Notably, the camphor and menthon show slightly abnormal effect on V_{oc} and FF. solar cells based on camphor-perovskite achieve the highest FF, and consequently, the top efficiency. This may also partially result from its smallest thickness. Because the FF in solar cell is closely related to the charge carrier transport feature in device, the thinner and more uniform reverse micelle layer is expected to be less insulative and thus be responsible for a better transport behavior among other biomolecule-treated films. Consequently, the highest PCE was observed in the camphor-perovskite device.

The variations observed in the thickness of the reverse micelle layer, despite utilizing biomolecules at consistent concentrations, could plausibly be ascribed to variations in the attractive forces exerted by the polar heads of the biomolecules towards the perovskite core within each reverse micelle nanoparticle. Influencing factors encompass steric hindrances, interactions between the biomolecules' nonpolar tails and solvent molecules, as well as the proximal effects of adjacent biomolecules tethered to the identical perovskite core. These elements, in concert, may induce divergences in the size, density, shape, and volume of the reverse micelle particles across different instances.

Fig. R27. (a) Cross-sectional SEM images for perovskite layer incorporated with different Tier 2 biomolecules. (b) Comparison of solar cell performance parameters with the top layer thickness.

11. It is hard to read the letters in Figure 5c. Please modify it.

Responses: We thank this comment. The **Fig. 5c** is modified accordingly.

12. In page 13, the correct sentence is as follows: Fig. 4c(i) shows the PL results of samples consisting of perovskite with a top Spiro-OMeTAD HTL.

Responses: We thank this comment. The sentence is corrected as suggested.

“... Fig. 4c(i) shows the PL results of samples consisting of perovskite with a top Spiro-OMeTAD HTL. ...”

13. Why the perovskite@bio-molecule nanoparticles have a larger bandgap.

Responses: We express our gratitude for this comment. As elucidated in **Fig. R25** (shown above), the electronic band structure of the perovskite@biomolecule nanoparticles undergo alterations following molecular interactions. There are potentially three hypothetical explanations for the observed enlargement in the bandgap:

(1) & (2) The attenuation of tail states near the band edge, as well as the diminution of deep states within the band, are apparent (as illustrated in **Fig. R25**). These phenomena are attributed to the modulation of the electron Pb's orbital induced by the biomolecule interaction.

(3) The third rationale might be associated with the quantum confinement effect, particularly noticeable due to the smaller dimensions of the nanoparticles. We have observed that the reverse micelle nanoparticles can exhibit dimensions as small as 10s of nm, analogous to those found in quantum dots. In such scenarios, the reduced size can lead to quantized electron orbitals, which in turn, contribute to the bandgap widening.

Together, these factors collaboratively account for the observed changes in the electronic properties of the perovskite@biomolecule nanoparticles.

REVIEWERS' COMMENTS

Reviewer #1 (Remarks to the Author):

The related questions have been solved. This work deserves publication in nature communications.

Reviewer #2 (Remarks to the Author):

Thank you for your detailed answers and thorough revisions. All my questions have been addressed.

Reviewer #3 (Remarks to the Author):

The manuscript has been fully revised based on the reviewers' comments, so I suggest its acceptance as it is.